# Placing the Common Era in a Holocene Context: Millennial-to-centennial patterns and trends in the hydroclimate of North America over the past 2000 years

Bryan N. Shuman[1], Cody Routson[2], Nicholas McKay[2], Sherilyn Fritz[3], Darrell Kaufman[2], Matthew E. Kirby[4], Connor Nolan[5], Gregory T. Pederson[6], and Jeannine-Marie St-Jacques[7]

[1] Roy J. Shlemon Center for Quaternary Studies, Department of Geology and Geophysics, University of Wyoming, Laramie, Wyoming, 82071, USA
[2] School of Earth Sciences & Environmental Sustainability, Northern Arizona University, Flagstaff, Arizona, 86011, USA
[3] Department of Earth and Atmospheric Sciences, University of Nebraska, Lincoln, Nebraska, 68588, USA
[4] Department of Geological Sciences, California State University, Fullerton, Fullerton, California 92834, USA
[5] Department of Geosciences, University of Arizona, Arizona, 85721, USA
[6] Northern Rocky Mountain Science Center, U.S. Geological Survey, Bozeman, Montana, 59715, USA
[7] Department of Geography, Planning and Environment, Concordia University, Montreal, Quebec, H3G 1M8, Canada

*Correspondence to*: Bryan N. Shuman (bshuman@uwyo.edu)

**Abstract.** A synthesis of 93 hydrologic records from across North and Central America, and adjacent tropical and Arctic islands, reveals centennial to millennial trends in the regional hydroclimates of the Common Era (CE; past 2000 years). The hydrological records derive from materials stored in lakes, bogs, caves, and ice from extant glaciers, which have the continuity through time to preserve low-frequency (>100 year) climate signals that may extend deeper into the Holocene. The most common pattern, represented in 46 (49%) of the records, indicates that the centuries before 1000 CE were drier than the centuries since that time. Principal components analysis indicates that millennial-scale trends represent the dominant pattern of variance in the southwest U.S., northeast U.S., the mid-continent, Pacific Northwest, the Arctic, and the tropics, although not all records within a region show the same direction of change. The Pacific Northwest and the southernmost tier of the tropical sites tended to dry toward present, as many other areas became wetter than before. In twenty-two records (24%), the Medieval period (800-1300 CE) was drier than the Little Ice Age (1400-1900 CE), but in many cases the difference was part of the longer millennial-scale trend, and, in 25 records (27%), the Medieval period represented a pluvial (wet) phase. Where quantitative records permitted a comparison, we found that centennial-scale fluctuations over the Common Era represented changes of 3-7% of the modern inter-annual range of variability in precipitation, but the accumulation of these long-term trends over the entirety of the Holocene caused recent centuries to be significantly wetter, on average, than most of the past 11,000 years.

## 1 Introduction

Hydroclimate extremes characterize the Common Era (CE; the past two millennia) across North America (Cook et al., 2007; Woodhouse and Overpeck, 1998) and other parts of the Northern Hemisphere (Ljungqvist et al., 2016). Wet and dry conditions shifted at annual to multi-decadal scales (Fye et al., 2003; Pederson et al., 2011; Woodhouse et al., 2009), with notable periods of "mega-drought." Mega-droughts are defined by severe moisture reductions lasting for multiple decades and covering large areas of the continent (Cook et al., 2004, 2010; Stahle et al., 2000). The Medieval period (800-1300 CE),

in particular, included several decades-long droughts that have been widely analyzed to assess the underlying mechanisms (Coats et al., 2014; Cook et al., 2004; Herweijer et al., 2007; Meko, 2007; Routson et al., 2016a; Seager et al., 2007a), although earlier portions of the Common Era may have been at least as dry (Routson et al., 2011). Throughout this interval, the observed hydroclimate changes had important consequences for people, landscapes, and ecosystems (Heusser et al., 2015; Jones and Schwitalla, 2008; Mason et al., 2004). As a result, the Common Era provides a useful baseline for

considering future hydroclimatic changes and their impacts (Cook et al., 2015; Seager et al., 2007b).

        Little is known, however, about the low-frequency trends that lasted for centuries or longer and slowly altered the hydroclimate of North America during the Common Era. Low-frequency trends may arise from externally forced trends extending throughout much of the Holocene as well as additional millennial to centennial variations. Trends persisting for centuries to millennia were probably small compared to inter-annual variability and may have fallen within the uncertainties

of reconstructions at annual scales (e.g., a 0.01 mm/yr multi-century trend in effective precipitation at Mina Lake, Minnesota over the past >800 years; St. Jacques et al., 2008, 2015). However, small changes that are imperceptible at annual scales can substantially affect hydrologic baselines when they accumulate over time scales of >100 years, reaching magnitudes at millennial scales equivalent to historic droughts, such as the 1930s Dust Bowl (Shuman et al., 2010).

        Tree-ring records have generated rich insight into annual to centennial-scale climate variability, but, with the

exception of a small number of chronologies comprised of long-lived individual series, most do not thoroughly preserve evidence of low-frequency (>100 year) changes (Ault et al., 2013; Cook et al., 1995; Sheppard et al., 1997). New methods have improved the extraction of low-frequency signals (Cook et al., 2010), but a comparison of low-frequency changes during the Common Era and earlier portions of the Holocene requires focusing on different types of records that extend further in time than most tree ring chronologies. The spatial coverage of long tree ring chronologies also differs from other

sources of information. Materials that have accumulated in lakes, bogs and caves, and ice stored in extant glaciers, can be used to evaluate centennial or longer trends, even if they often lack annual-scale age control or resolution (Marlon et al., 2016). The strength of these longer-lived archives resides in the continuous, multi-millennial time series; they have been studied in regions, like the northeastern U.S., where tree ring records may only be viable for a few hundred years.

Weaknesses often derive from lack of annual-scale age control and sample resolution, which can further limit calibration in time against observed time series of historic hydroclimate changes (although calibration of 100-1000 year signals is difficult anyway because of the absence of long historic records). These limitations have not, however, prevented many records from revealing a rich record of low-frequency climate changes during the Holocene, which have often been corroborated by data

from multiple sites and methods (e.g., Hodell et al., 2005b; Marlon et al., 2017; Shuman and Marsicek, 2016). We focus on these approaches to place Common Era trends and events in a Holocene context. Because the hydroclimate sensitivities involved may also differ from those of tree rings, new insights into data interpretation may ultimately arise from comparison with dendroclimatic reconstructions, such as the North America Drought Atlas (Cook et al., 2013, 2010).

      Here, we present a dataset of 93 records of hydroclimate change during the Common Era in North America, Central

America, and the adjacent islands from the tropics to the Arctic (Fig. 1). All the records are published and span >1000 years. Many approach decadal-scale resolution, but they all have a mean resolution of at least one sample per century. The records represent a wide range of different physical, biotic, and geochemical variables. Because they derive from multiple archives and measurable variables, they record different characteristics of local hydrological changes and yield a diverse matrix of information about factors ranging from lake levels and soil moisture to evaporation (Table 1). Some may be sensitive to the

frequency of annual events (e.g., droughts), but others represent long-term mean conditions via autoregressive processes such as groundwater integration times. Annual precipitation has been inferred from 14 fossil pollen records (15% of the records), but most other records (58 or 62%) have been interpreted to preserve evidence of changes in effective moisture (precipitation minus evaporation). Twenty-five records (27%) provide calibrated reconstructions of hydroclimatic variables (e.g., millimeters of precipitation), but most represent relative indices of hydrologic change. To simplify treatment of this

diverse data matrix, we discuss the patterns in the data in terms of relative changes from wet to dry conditions, broadly defined.

      We use this dataset to identify the major trends, if any, that characterize the hydroclimate of North America over the past two millennia. Using data binned by century, we evaluate the patterns associated with commonly referenced periods, such as the Medieval Climate Anomaly (MCA) from 800–1300 CE (1150-650 BP) and the Little Ice Age (LIA) from 1400-

1900 CE (550-50 BP), but we also describe other first-order hydroclimate trends represented by the data. In doing so, we assess evidence for major regional contrasts that may represent important circulation patterns or dynamics (e.g., Mock, 1996; Shinker and Bartlein, 2010; Wise and Dannenberg, 2014), and, where possible, we consider the magnitude of Common Era trends in the context of the Holocene.

## 2. Data and Methods

### 2.1. Dataset

### 2.1.1 Overview

The 93 lake, bog, cave, and glacial records span nine different regions of North and Central America, Greenland, and the
Caribbean islands, with variable spatial densities of sites among regions (Fig. 1, Table 1). The nine regions were determined before we compiled the data considered here (McKay, 2014), and were based on the level 1 ecoregions of North America (Commission for Environmental Cooperation Working Group, 1997) and major patterns of covariance within modern climate data (Mock, 1996). Where spatial outliers existed separate from the main cluster of data within a region (e.g., data from Florida versus the northeast U.S.; northern versus southern Great Plains), we split our initial regions to ensure suitable
representation of the data in our analysis. We used these designations to ask whether distinct trends were recognizable among commonly recognized regions and whether any trends have parallels to patterns of climate variation observed at finer time scales, such as north-south anti-phased moisture variability along the western margin of North America (Cayan, 1996; Wise and Dannenberg, 2014). To facilitate comparison of centennial and longer trends, we calculated mean values for each record per century.

15        Most of the records have been interpreted to represent a specific hydrologic variable, such as annual precipitation, lake or bog water depth, lake salinity, snow accumulation rate, or represent a geochemical index related to hydrology, such as oxygen and carbon stable isotope concentrations or other elemental ratios reflective of evaporative concentration of ions in soil or lake water. We have omitted any fossil datasets (e.g., pollen or diatom percentages) comprised only of relative or qualitative indices, and only 9 out of 93 records represent relative indices based on sedimentological properties related to
processes such as runoff or dust deposition. In two cases, Deep Pond, Massachusetts and Castor Lake, Washington, we included two reconstructions based on different independent analyses or cores, but most sites are represented by a single time series. To focus on multi-century trends in the mean hydroclimate, we also have excluded sedimentary records of extreme events, such as hurricanes or floods (Donnelly et al., 2015; Munoz et al., 2015).

        We compiled the data from publically available sources and solicited additional published records from the original
authors. Many of the records derive from NOAA's Paleoclimatology database (https://www.ncdc.noaa.gov/data-access/paleoclimatology-data/datasets) where our dataset can also be accessed (https://www.ncdc.noaa.gov/paleo/study/22732).

### 2.1.2. Lake datasets: biotic, geochemical, sedimentological

The vast majority of records (77%) come from lake sediments, which are available from all regions except the southern Plains and southeastern U.S. (Fig. 1, Table 1). These records represent a range of different approaches to hydrologic

reconstruction. Fossil pollen data have been used to infer past precipitation changes, and we include published reconstructions where they are site specific rather than regional averages (e.g., Gajewski, 1988). Likewise, we also include records of paleolimnological status inferred from transfer functions applied to fossil diatoms and other microfossil assemblages. These inferences include variables sensitive to moisture balance, such as water salinity and water depth (e.g., Laird et al., 1996; Schmieder et al., 2011).

In addition to the biotic records, we include sedimentological datasets, which typically are used to infer changes in lake depth (Fig. 1). In some cases, the sedimentological data have been converted to quantitative estimates of lake level (e.g., Pribyl and Shuman, 2014), which, in a subset of these records, have been used to calculate changes in precipitation minus evaporation (e.g., Marsicek et al., 2013). In others, the data provide a relative index of lake status. In a few cases, lake sedimentology has also been used as a relative measure of precipitation-related run-off and aeolian activity (Dean, 1997; Kirby et al., 2010; Routson et al., 2016b).

Finally, we include geochemical records, focusing on the original authors' interpretation of the analyses. The geochemical records range from sulfur concentrations in lake sediments indicative of evaporative concentration and associated water-level change (Hodell et al., 2001) to Mg/Ca and oxygen isotope analyses of lake carbonates indicative of evaporative losses from the lakes (Tian et al., 2006; Yu and Ito, 1999). Oxygen isotope datasets from lakes also have been used to infer changes in precipitation seasonality or source (e.g., importance of the local snowpack versus direct precipitation)(e.g., Anderson, 2011; Steinman et al., 2012). Other compounding factors also influence these records (Anderson et al., 2016; Steinman and Abbott, 2013), but most of the lake sediment isotope records have been used as relative indices of high versus low effective moisture. However, to focus on the patterns of local hydrologic changes, we excluded records thought to have been dominated by local temperature effects or large-scale circulation changes rather than their local hydrologic effects (e.g., Anderson et al., 2005).

### 2.1.3 Peat sequences

In raised ombrotrophic bogs in Minnesota, Michigan, Maine, and Quebec, variability in precipitation and evaporation changes the surface wetness of the bog and, thus, the water-table depth, which influences the community composition of testate amoebae (Booth, 2010). Like biotic indicators from lake sediments, testate amoebae leave identifiable remains, which can be used along with extensive networks of modern samples to reconstruct past water-table depths (Booth, 2008). Water-table-depth reconstructions are often detrended to minimize the influence of autogenic bog processes and to emphasize decadal-to-centennial-scale events, attributable to climate rather than wetland evolution (Booth, 2010). Other evidence of hydrologic change can be derived from peatlands, but only calibrated estimates of water table depth from testate amoebae records are included here.

### 2.1.4 Speleothems and other cave deposits

Sediments and carbonate speleothems from caves also preserve oxygen and carbon isotopes, and trace-element chemistry (Mg/Ca, Sr/Ca), which can be used to infer past hydroclimates. In our data compilation, we include eight of these records from Central America and scattered sites at latitudes <42° N in North America. Like oxygen isotopes from lake sediments, many different processes can affect these records, but we followed the interpretations of the original publications.

### 2.1.5 Glacier ice

We include four annual ice-layer-accumulation records from the Greenland ice sheet (Meese et al., 1994; Andersen et al., 2006). These cores were dated by counting annual layers as determined from visual stratigraphy, laser light scattering from dust, chemistry, and electrical conductivity, with cross-checking using known volcanic eruptions. These accumulation records were then corrected for densification and ice-layer thinning due to ice flow, and we use them as direct measure of past precipitation.

### 2.2. Statistical Analyses

### 2.2.1 Binning, normalization and infilling

To compare the time series from these disparate sources, we averaged them into discrete bins and standardized them over the Common Era. The bins were calculated for each record at 100-yr intervals, averaging all the points within each century. Binning emphasizes long trends and removes age-uncertain sub-centennial variations from each record. We standardized each binned record over the 2000-year interval of the Common Era, or the entire length of the record if shorter than the analysis period. To do so, we subtracted each record's mean and divided by its standard deviation to calculate z-scores. Bins with missing data were infilled for cluster and empirical orthogonal function and principal components (EOF/PCA) analyses using singular spectrum analysis (SSA) to estimate the values of missing data (Ghil et al., 2002). SSA is commonly used to iteratively infill dataset gaps by relying on the spatiotemporal covariance within the dataset. This methodology is well suited for infilling gaps in bins with no observations, but is not well suited for extrapolation. Missing bins were infilled with SSA using a 5-bin (500yr) moving window.

### 2.2.2 Cluster analyses

Hierarchical cluster analysis (HCA) was used to organize the records into a binary tree that subdivides groups of records with similar temporal patterns (Bar-Joseph et al., 2001; Eisen et al., 1998). Cluster analysis has been widely applied in paleoecology to divide samples into stratigraphic zones (Grimm, 1987), but we use HCA to group paleoclimate datasets by their dominant temporal patterns (e.g., Kaufman et al., 2016). The HCA and associated heat map was computed with the clustergram.m program in Matlab, using Euclidian pairwise distances between records for clustering and average linkages for the dendrogram. The composites were then generated by averaging the binned time series from each cluster. The 95% bootstrap confidence intervals on the composites were generated over 500 replicates of sampling-with-replacement from the records contributing to each cluster (Boos, 2003).

### 2.2.3 EOF/PCA

Principal component analysis (PCA) and empirical orthogonal function (EOF) analysis were used to characterize the dominant modes of variability in the North American hydroclimate dataset, as well as to characterize regional patterns of variability. The PCA was conducted on regional subsets of sites, based on regions represented by polygons in Figure 1, and was applied using base functions in R (R Core Development Team, 2009). The PCA-by-region analysis was conducted to evaluate the strength of any signals within each region with >10 records, rather than simply calculating mean trends, and to assess potential correlations or shared signals across geographically distinct regions. The EOF analysis evaluates the latter from the perspective of the whole dataset.

### 2.2.4 Epoch difference and trend analyses

Epoch differences were used to map broad-scale changes in the hydrological inferences between time periods. We used this method to examine two pairs of time periods: 1) the MCA 800–1300 CE minus the LIA 1400-1900 CE, and 2) the second millennium (1000-2000 CE) minus the first millennium (1-1000 CE). The latter analysis of the difference between the two millennia was designed to evaluate the spatial patterns of any long-term trends and to assess whether any of the MCA-LIA differences were a function of longer trends. We compare the epoch differences with those calculated using the North American Drought Atlas (Cook and Krusic 2008). Additionally, we fit generalized linear models to records that have been calibrated to provide quantitative hydroclimate time series to determine the magnitude of trends over the Common Era. The models were fit to binned data while accounting for correlation in the error terms using a first-order autoregressive model in the nlme package in R (Pinheiro et al., 2017).

# 3. Results

## 3.1 Temporal patterns

A heat map shows the individual records and their departures from the mean of the Common Era (Fig. 2A). Cluster analyses of these data identify three major groups that differ in their long-term trends (Fig. 2B-D). The averaged time series of the records (n = 25) included in the first of the clusters, indicated in green in Figure 2, indicates wet conditions from 800-1500 CE (including during the MCA) followed by drying to present (Fig. 2B). The wet phases at individual sites varied in time, but most often occurred from 800-1200 CE (green in Fig. 2A, top). This cluster primarily includes sites in the Great Lakes region, the northeastern U.S., and adjacent areas (Fig. 2E). Because few data exist for some regions, such as the southeast U.S. (Fig. 1), the absence of any patterns may be a function of data availability.

The second cluster, shown in blue in Figure 2, includes records (n = 22) that are distinguished for showing pronounced evidence of drought from 900-1300 CE during the MCA (Fig. 2C). Many of these data also have long-term drying trends toward present and indicate wet conditions early in the Common Era before 500 CE, which are not repeated after the MCA (Fig. 2A). Overall, the sites that comprise this cluster are broadly distributed. It includes the majority of records in the Arctic and boreal Canada, but these patterns are also characteristic of a few sites in western North America (Fig. 2F).

The third cluster, which is marked as red in Figure 2, includes sites with a prevailing trend from dry to wet over the Common Era (Fig. 2D). Excluding the detrended bog records, which by definition cannot capture millennial-scale changes, this cluster represents 53% (46 of 87 records) of our dataset. The 46 records include the majority of records in the northeast U.S., the mid-continent, the Rocky Mountains, and the Yucatan Peninsula, as well as along the western margin of North America (Fig. 2G).

Breaking out the data by archive type further reveals that the lake records, which dominate the dataset, have a mean moistening trend over the Common Era equal to approximately 1 standard deviation (Fig. 3). The means of other archives represent fewer records, and thus, retain similar magnitude (1 standard deviation) variations at centennial scales. Despite these short-lived variations, often-studied periods such as the MCA and LIA are not distinct in most of the mean time series. Instead, ice cores (representing Greenland) and speleothems (dominantly representing western North America and central America) record mean drying trends also equaling about 0.5-1 standard deviations (Fig. 3).

## 3.2 EOF results

The pattern and interpretation of the first EOF indicates that overall millennial-scale trends, particularly those amplified by a transition at ca. 900-1200 CE, explain the largest amount of coherent variance in the data (Fig. 4A-B). This pattern is similar to that of the third cluster. The first EOF loads positively and most strongly at sites in the northeast U.S., the western mid-continent, the western margin of North America, and the Yucatan Peninsula. It also loads negatively in many regions (Fig.

4A), but few individual records show strong drying trends to present (Fig. 2A), and the weak or negative loadings appear to primarily indicate records with little or no trend. Proportionally, however, the variance explained by EOF 1 is small (29.4%).

The second EOF emphasizes contrasts between ca. 1000 CE and the last few centuries, and accounts for 18.1% of variance in the data (Fig. 4C-D). Like clusters 1 and 2, EOF 2 highlights the Medieval-age anomalies in the records, but they neither dominate the variability within the dataset nor are they consistently of the same sign. The EOF loads negatively (dry at ca. 1000 CE and wet today) across large areas of western and tropical North America, whereas positive loadings (wet at ca. 1000 CE and dry today) appear most commonly in the northeast U.S. and in Greenland, consistent with the pattern of sites that comprise cluster 1 (Fig. 2B).

The third EOF is the only other EOF to explain >10% of the variance (11.8% explained, Fig. 4E-F), and it highlights sites where the 19th and 20th centuries contrast with earlier periods (like the first cluster in Fig. 2A). It loads most heavily and positively (dry today, wet LIA) in the northeast U.S., central Rocky Mountains, and Central America and has negative loadings (wet today, dry LIA) across northwestern North America from Washington to Alaska and Manitoba (Fig. 4E).

Overall, the total variance explained by the first three EOFs (59.3%) is slightly more than half, which indicates that no one pattern consistently dominates the dataset. Thus, more than half of the variance attributable to these dominant EOFs is associated with synoptic-scale trends (EOF 1), but nearly half of the total variation also can be attributed to site-specific changes and heterogeneity within regions (Fig. 2A).

### 3.3 Epoch differences

A map of the differences between the mean z-scores for the LIA (1400-1900 CE) and the MCA (800-1300 CE) highlights the extent of low-frequency drying during the MCA (Fig. 5A), which is not consistently the same as the patterns of individual annual to multi-decadal droughts during the same time (Herweijer et al., 2007). Red symbols on the map represent a dry MCA: sites with higher z-scores (wetter) during the LIA than during the MCA. Reduced moisture during the MCA extended from southern California through the central Rocky Mountains to areas of the mid-continent (Fig. 5A). Dry conditions also affected the northeast U.S. and Yucatan, but the difference between 800-1300 CE and 1400-1900 CE at many sites simply reflects the millennial-scale wetting trend over the Common Era (Fig. 2D). Blue symbols indicate that some areas, such as in the Pacific Northwest, were also wet during the MCA (Fig. 5A).

A map contrasting the first and second millennia CE (Fig. 5B) shares many features of the LIA-MCA difference map and the patterns of EOF 1, which emphasize the long-term trends (Fig. 4A-B). Red symbols indicate where the first millennium CE was drier than the second, and blue symbols show the reverse (where the mean z-scores for the second millennium CE were higher, wetter, than those for the first millennium). A wetter first millennium CE was most prominent in the Pacific Northwest and in southern Central America (blue, Fig. 5B), whereas the first millennium was drier over portions of the Yucatan, California, the Rocky Mountains, the mid-continent, and the northeast U.S. (red, Fig. 5B).

The greatest difference between the epoch difference maps (Fig. 5A versus Fig. 5B) exists in the mid-continent and Rocky Mountains. Histograms of the total numbers of wet and dry sites both skew positively (toward wet conditions in the more recent intervals), but the LIA-MCA is skewed more toward a dry MCA than the millennial difference, which includes a larger proportion of sites recording a wet first millennium (Fig. 5B, inset). Many anomalies also cluster more coherently (e.g., in the Pacific Northwest) in the LIA-MCA difference map (Fig. 5A) than the millennial difference map (Fig. 5B).

The patterns are not consistently well correlated with the millennial mean differences in Palmer Drought Severity Index derived from the North American Drought Atlas dendroclimatic dataset (Fig. 5). The best agreement exists in western North America where both datasets show that the MCA was wetter than the LIA in much of the Pacific Northwest, but drier than the LIA across the southwest (Fig. 5A). Agreement is poor, however, in mid-continent and eastern areas. The agreement is also low when comparing the mean of the two millennia (Fig. 5B).

### 3.4 Regional correlations and contrasts

Time series of first principal component (PC) scores, derived from the records within each region in Fig. 1, indicate that similar long-term changes affected multiple regions (Fig. 6). In particular, many regional PC scores represent large differences between the two millennia and show evidence of a rapid transition at ca. 1000 CE. These differences in time explain about a third of the variance in most regions (27-41%, Fig. 6).

The regional scores are correlated (absolute magnitude of r > 0.55, p ≤ 0.01), but in most cases, the correlations only represent long-term autocorrelated trends. Based on the maps (Fig. 5) and loadings, the changes represented by the correlated scores generally trended in the opposite direction in the Pacific Northwest, the Arctic and parts of the tropics compared to the other regions. The Pacific Northwest tended to become drier toward present as large portions of the Southwest, northeast U.S., and the mid-continent became wetter than during the first millennium CE (Fig. 5B). Generalized least squares models including a first-order autoregressive term indicate that the PC scores only correlate significantly between the Southwest and two other regions, the northeast U.S. (slope, β = 0.68±0.13, p = 0.0001) and the Pacific Northwest (β = 0.38±0.14, p = 0.0193). The differences among the geographic regions also parallel the geographic biases in the different archive types (Fig. 3).

### 3.5 Holocene context

A subset of Common Era records extends at least 6000 years into the Holocene and has a similar geographic distribution as the overall Common Era dataset. The Holocene records indicate that the dominant wetting trend over the past 2000 years began early in the Holocene (Fig. 7). Maps reveal that individual regions, such as portions of northwestern and tropical North America, did not always follow the mean pattern, but the Holocene-length records facilitate comparisons of the mean trends for both time scales (Fig. 7A). On average, the Holocene-long records (n = 36) show a long-term increase in

moisture leading to the Common Era (Fig. 7B). The period from 1000-1500 CE, specifically, was the wettest of the Holocene based on this set of data. By contrast, the average z-scores were 5 standard deviations below the mean of the Common Era during the early Holocene.

**3.6 Magnitudes of change**

Moisture-balance changes inferred from past lake levels, pollen-inferred annual precipitation, and ice accumulation rates provide constraints on the magnitudes of the low-frequency hydroclimate changes. Such data are spatially clustered and as such may not be representative of the whole continent. However, where they exist, they reveal the magnitude of the fluctuations described above (Fig. 8). Overall, the reconstructions indicate changes equivalent to net changes in precipitation

on the order of 0-100 mm over the Common Era. Negative trends (red, Fig. 8A) are uncommon in this subset of the records and have smaller magnitudes than the positive trends (blue, Fig. 8A). Consequently, the inferred rates of change before 1500 CE (before potential land use effects on the pollen records) equal -0.01 to +0.07 mm/yr (Fig. 8B).

In the northeast U.S., where the greatest density of quantitative records exists, the lake-level and pollen records indicate an increase in annual effective precipitation of 25-50 mm, about 2-5% of mean annual precipitation in the region

today, since the early part of the first millennium CE (Gajewski, 1988; Marsicek et al., 2013; Newby et al., 2014). No distinct differences exist between the pollen-inferred precipitation changes and changes estimated from lake levels (Marlon et al., 2017; Marsicek et al., 2013). Trends range from having magnitudes similar to those in northeast to no significant trends in pollen-inferred precipitation reconstructions from the Great Lakes region (Gajewski, 1988; St. Jacques et al., 2008). In the Rocky Mountains, lake-level reconstructions from Wyoming also indicate an increase of ~20 mm (4%) since 100 CE,

when conditions were approximately as dry as during the MCA (Pribyl and Shuman, 2014). The changes in the northeast U.S. and Wyoming equal about 3-7% of the range of inter-annual variability in annual precipitation, as represented by climate division data since 1948 CE (NCDC, 1994). Trends in Greenland ice core accumulation rates have magnitudes near zero; the largest and most significant trend equals -0.005±0.002 mm/yr (p = 0.07) at GRIP (Andersen et al., 2006).

**4. Discussion**

**4.1 Prominent low-frequency patterns**

Multi-century fluctuations and trends appear to be important features of the hydroclimatic history of North America during the Common Era. The causes and consequences of these low-frequency (centennial to millennial scale) hydrologic changes need to be considered as part of the full spectrum of changes during recent millennia, in addition to the annual to multi-decadal variability that has been the focus of other syntheses of Common Era climate patterns (Ault et al., 2013). An overall

trend from dry to wet dominates >50% of the records (Fig. 2) and is evident in the first EOF, which explains 29.4% of the variance in the available dataset of 93 records (Fig. 4A-B). Additional multi-century patterns, represented by clusters 1-2 and

EOFs 2-3, also express coherence across sets of records and multiple regions and indicate that long wet periods were as frequently recorded as prolonged dry intervals, such as the MCA (Fig. 2A-B). Overall, periods like the MCA were not unusual, because hydroclimate fluctuations created periods that were both wetter and drier than present, although few centuries appear to have been as wet as the 20[th] century.

5          The spatial heterogeneity in the temporal patterns (Fig. 4-6) may represent real sub-regional complexity of hydroclimate, consistent with the variability associated with historic hydroclimatic changes (Cook et al., 2007; Dai et al., 1998; Groisman and Easterling, 1994). For example, maps of EOF1 (Fig. 4A) and the epoch differences (Fig. 5) highlight a contrast between the Pacific Northwest and much of the southwest and mid-continent as well as other areas such as the northern Yucatan peninsula and the northeast U.S. The clustering of anomalies in western North America is well defined for

the LIA-MCA difference maps (Fig. 5A) and has parallels in latitudinal shifts in moisture anomalies at annual to decadal scales, such as associated with changes in Pacific sea-surface temperature patterns today (Dettinger et al., 1998; Wise, 2010, 2016). The LIA-MCA pattern includes some similarities to maps of the frequency of drought during the Dust Bowl era from 1924-1943 CE, which included low frequencies of drought in the Pacific Northwest when drought was frequent in other regions (McCabe et al., 2004). Likewise, the "Terminal Classic Drought" in the Yucatan from ca. 700-1100 CE may include

both low-frequency trends and frequent decadal-scale events (Hodell et al., 2005b; Lachniet et al., 2012; Medina-Elizalde et al., 2010), but the limited spatial extent of the drought signal (Fig. 5) corresponds closely to the focused area of well-correlated precipitation in the Yucatan today (Medina-Elizalde et al., 2010). Additional heterogeneity evident in the maps may represent spatial complexity in hydroclimatic variables, such as the ratio of actual to potential evapotranspiration, the seasonal timing of maximum precipitation, and magnitude of monthly-scale changes in precipitation, which today can

include major differences across distances of ~100 km particularly through interactions with factors like topography (Shinker, 2010; Shinker and Bartlein, 2010).

         Non-hydroclimatic influences on the proxy records and site-level factors also may explain much of the 40.7% of the variance unattributed to EOFs 1-3, and these influences are also likely embedded in the first three EOFs as well. Differences among archives, such as their sensitivities to annual versus seasonal hydroclimate, also may explain why Medieval drying

from 900-1300 CE (1150-650 BP) can appear in one sub-set of records from a given region (Fig. 2C), while long trends from dry to wet dominate another sub-set from the same area (Fig. 2D). One potential implication is that long-term trends (>100-1000 years) affected more of North America than is apparent from our dataset, but some archives (e.g., detrended bog records) do not preserve such signals. Some records may act as stringent low-pass filters (e.g., sedimentary records of lake-level change; Newby et al., 2014), but others may be better suited to recording discrete events (e.g., bogs in the northeast

U.S.; Booth et al., 2006). Local hydrological factors may also be important for creating differences from site to site, or counter-intuitive shifts in variables like oxygen isotopes, particularly in lakes, even when a region experiences an extreme drought (Donovan et al., 2002; Fritz et al., 2000; Plank and Shuman, 2009).

         Furthermore, the climate signals were weak relative to our ability to reconstruct them. When considered over the entire Holocene, the data show pronounced trends in hydroclimate that far exceed those of the Common Era (Fig. 7), but, at

the scale of individual millennia or centuries such as those of the last 2000 years, the signal-to-noise ratio in many records may be relatively low. For example, independent lake-level and pollen-derived estimates of Common Era changes can correlate closely (Marlon et al., 2017; Marsicek et al., 2013), but the root mean squared error of pollen-inferred annual precipitation equals ~165 mm compared to trends of <50 mm over 2000 years (Fig. 8).

5        The small magnitudes of past hydroclimate fluctuations reveal, however, the unusual character of the changes taking place in some regions today. For example, in Massachusetts in the northeast U.S., multiple pollen and lake-level datasets indicate that effective annual precipitation increased by <50 mm over 2000 years or <0.025 mm/yr (Fig. 8B)(Marsicek et al., 2013). By comparison, annual precipitation in that area has increased by 2.7 mm/yr since 1948 CE (NCDC, 1994; Pederson et al., 2013). More generally, annual precipitation increases of 13% over southern Canada and 4%

over the U.S. during the 20[th] century (Groisman and Easterling, 1994) could represent at least an order-of-magnitude amplification of the regional trends of <5% during the entire Common Era (Fig. 8). Additional work is required, however, to thoroughly compare the magnitudes and rates of past and ongoing trends.

### 4.2 Medieval drying and low-frequency records

Our data confirm that during the MCA, average moisture decreased over large areas of North America (Fig. 2C, 6). However, at many sites, the MCA was drier than the LIA or other more recent periods, simply because of the long-term trend, such that earlier periods tended to be drier than later periods (e.g., the red cluster 3 in Fig. 2D; EOF 1, Fig. 4A-B). As a result, the dry periods in the early portions of some dendroclimatic records that extend back <1000 years might capture only part of a trend that would have extended further back in time (e.g., Pederson et al., 2013). Comparison of the epoch

difference maps for the MCA-LIA (Fig. 5A) and the first-second millennia (Fig. 5B) highlights that MCA drying (red circles in Fig. 5A) was more coherent and extensive in the mid-continent and southwest U.S. than the mean aridity of the first millennium (red circles in Fig. 5B). Indeed, many mid-continent and southwestern records indicate that the first millennium was wetter than the second (blue circles in Fig. 5B). Thus, the mid-continent best expresses a distinct MCA dry anomaly (blue cluster 2 in Fig. 2C,F), but also evidence of millennial-scale moistening (the red cluster 3 in Fig. 2D, G).

25        The 25 records included in cluster 1 (green, Figure 2B) also indicate that unusually wet conditions characterized some regions during the MCA (Fig. 2A-B), which is consistent with several multi-decadal pluvials in dendroclimatic reconstructions in western North America (Routson et al., 2016a). Furthermore, long drying trends in regions, such as the Pacific Northwest, resulted in a LIA that was drier than the MCA (Fig. 5-6). However, in the northeast U.S., where many records fall into cluster 1 (Fig. 2E), the pollen assemblages that provide much of the evidence for the inferred drying since

the MCA are influenced by land use since ca. 1600 CE and should be interpreted cautiously (Fuller et al., 1998; Marsicek et al., 2013; St. Jacques et al., 2015; Webb et al., 1993). The importance of ragweed (*Ambrosia*) and other herb pollen in the upper portions of these records result in modern analogs that derive from drier prairie regions, which may explain why pollen-inferred precipitation reconstructions deviate significantly from lake-level reconstructions (within cluster 3, Fig. 2D)

only during the LIA (Marsicek et al., 2013). If so, the land-use effects on the pollen-derived reconstructions may artificially truncate multi-millennial moistening trends otherwise found in the northeast U.S. (Fig. 2G). Nonetheless, anomalously wet regional conditions during the MCA highlight the complex nature of hydroclimate during this interval in general.

## 4.3 Differences between the first and second millennia

The long trends apparent in our dataset typically indicate that the mean state of the first and second millennia differed (Fig. 5B) and that low-frequency trends differed between regions (Fig. 6). These contrasts have parallels to those observed in instrumental and dendroclimatic data, particularly in the western U.S. where the Pacific Northwest and southwest U.S. often exhibit anti-phased moisture anomalies, because of factors, such as tropical and northern Pacific sea-surface temperatures, which can influence the position of the jet stream (Dettinger et al., 1998; Routson et al., 2016a; Wise, 2010, 2016). As noted above, the contrast between the Pacific Northwest and the southwest U.S. bears similarities to patterns of historic drought (Cook et al., 2007; McCabe et al., 2004). The differences could indicate a tendency for storms to follow more southern tracks since 1000 CE than before, which resulted in general drying of the Northwest, likely due to a reduction in precipitation rather than a negative evapotranspiration-driven effect from temperature (PAGES 2k Consortium, 2013; Trouet et al., 2013; Wahl et al., 2012).

Central America represents another region with coherent signals of millennial change. Evidence of drought from ca. 700-1100 CE during the "Terminal Classic Drought" is evident in the Yucatan (Hodell et al., 2005b; Lachniet et al., 2012; Medina-Elizalde et al., 2010) and contributes to the difference between millennia captured by the third cluster of records (red in Fig. 2), but some records from the Yucatan also exhibit an increase in mean moisture levels from the first to second millennium beyond the specific droughts (Curtis et al., 1996). The pattern contrasts with the direction of change further south in Guatemala, Nicaragua, and Panama where lake and speleothem oxygen isotope records indicate that moisture levels fell from the first to second millennium (Fig. 5)(Curtis et al., 1998; Lachniet, 2004; Rosenmeier et al., 2002; Stansell et al., 2013). The pattern provides support for a zonal shift in tropical precipitation across the region, although at least portions of the Yucatan also became drier than before during the LIA (Hodell et al., 2005a). The changes could have resulted from factors such as the temporal evolution of the El Nino-Southern Oscillation (ENSO), dynamical processes over the Atlantic, or changes in the latitudinal temperature gradient that influenced the position of the American monsoon or inter-tropical convergence zone.

The northeast U.S. and adjacent Canada represent a third region where coherent millennial differences exist among records (Fig. 5). Many pollen and lake-level records indicate that the region became increasingly wet over the last 2000 years (Marlon et al., 2017; Marsicek et al., 2013; Newby et al., 2014). Regional cooling may play an important role (Shuman and Marsicek, 2016), and if so, different regional patterns could have resulted from different processes. Combinations of atmospheric circulation or dynamical changes, such as zonal precipitation shifts, may have been important in western North

America and central America, while the influence of direct energy budget changes on factors like evaporation or snowmelt could have influenced patterns in other regions like the northeast U.S.

Placed in the context of the Holocene (Fig. 7), the difference between moisture levels of the two millennia of the Common Era is modest and consistent with longer-term multi-millennial trends. Changes in global forcing, such as slow changes in seasonal insolation and greenhouse gases, can explain many of the Holocene-scale trends (Shuman and Marsicek, 2016). Climate models run with mid-Holocene (4050 BCE or 6000 YBP) forcing simulate extensive drying; thus, like empirical data, the models indicate moistening towards the present, which was caused by dynamical responses of the atmosphere to seasonal insolation change, sea-surface conditions in the tropical Pacific and Atlantic, and surface-atmosphere feedbacks (Diffenbaugh et al., 2006; Harrison et al., 2003; Shin et al., 2006). Because of the responses to external forcing, the Common Era and especially the second millennium CE stand out as the wettest periods of the Holocene over much of North America, despite the decadal "mega-droughts" that occasionally punctuated the persistence of the relatively wet conditions. Understanding how the multi-centennially trends, which are expressed weakly at annual scales, interacted with such extremes at decadal scales warrants further investigation.

## 4.4 Comparison with dendroclimate records

The patterns here differ from those in the North American Drought Atlas (NADA, Cook and Krusic 2008), although the MCA-LIA patterns in western North America correspond closely (Fig. 5A). The differences elsewhere, and with regard to the longer-term millennial mean differences (Fig. 5B), may exist for several reasons. First, contrasts may exist between the way our dataset retains signals of annual-to-decadal variations (clearly preserved in the dendroclimate record) and those of multi-century and longer variations. For example slow sediment dynamics or forest tree longevity may prove resilient to annual variations, but readily responsive to change over centuries. Different map patterns could arise, therefore, because the 2008 version of the NADA (used in Fig. 5) likely emphasizes interannual variation, even when smoothed over centuries, whereas other datasets may emphasize the effects of centennial and longer changes. The differences would represent different patterns in the averages of high-frequency variability versus the patterns in low-frequency trends.

Furthermore, our dataset may lack a consistent ability to detect either annual-decadal variability or multi-century trends in a limited 2000-yr window because of the interaction of taphonomic process (e.g., sediment mixing) and the small magnitudes of the low-frequency trends. Some datasets may well be noisy relative to weak low-frequency signals. As we noted in the introduction, the magnitudes of the trends in many records are small over even 2000 yrs (Fig. 8) when compared to many reconstruction uncertainties. The low signal-to-noise ratio may also apply to the dendroclimate data at multi-century to millennial scales, and without long observational datasets available for validation, it is difficult to assess.

A third related explanation for mismatches could be that dendroclimatic reconstructions of variables such as the Palmer Drought Severity Index (PDSI) may differ from the hydroclimate variables represented by our data (e.g., net snow accumulation in ice cores; P-ET that drives lake-level changes). We do recognize some clusters of coherent anomalies (e.g.,

clusters of opposite sign anomalies in the Pacific Northwest versus the U.S. Southwest in Fig. 5), which would at first pass, suggest real signal and thus, in part, require explanations that involve differences in the time scale (explanation 1) or controlling variable (explanation 3) recorded by the two different datasets. More work is needed to test the various explanations.

## 5. Conclusions

Hydrologic reconstructions from across North and Central America, and adjacent islands, indicate that the hydroclimates of the first and second millennia CE differed significantly. Many regions, such as the northeast U.S., the mid-continent, the Rocky Mountains, the western margin of North America, and the northern Yucatan Peninsula experienced an overall wetting trend over the Common Era. Not all portions of North America experienced this trend, however, and the Pacific Northwest

and the southern tropics, in particular, were characterized by drying toward present. In many areas, the changes continue trends that persisted for millennia during the Holocene, and the patterns may indicate the continuation of long-term, low-frequency responses of Hadley circulation, the position and influence of the subtropical high systems, and the major westerly storm tracks to orbital and greenhouse gas forcing (Braconnot et al., 2007; Harrison et al., 2003).

The millennial-scale trends equal changes of <5% in annual precipitation over the Common Era and only 3-7% of

the historic range of inter-annual precipitation variability. However, they represent trends that allowed the Common Era to become the wettest portion of the Holocene in many areas. Multi-decade "mega-droughts" of the MCA were associated with low-frequency (>100-yr) departures from more recent wet conditions, but the low moisture levels of the MCA were commonly part of the long-term trends that made early periods drier than later portions of the Common Era. Now, however, increases in precipitation in many portions of North America have substantially accelerated the long-term trends.

**Data availability**

*Note to editor/reviewers: 43 datasets used in this study have not previously been archived. We are currently facilitating the transfer of the original data from the data generators to NOAA Paleoclimatology. Each dataset will receive a unique URL, which will be added to Table 1 prior to the final acceptance of this paper. At that time, we will also transfer the output time*

*series to NOAA. We will contact the editor as soon as the final datasets have been archived.*

(1) Input data used for the analysis: Table 1 includes the persistent online identifiers for the 93 original datasets used in this synthesis, which are available through the World Data Service (NOAA Paleoclimatology) or other public repositories. In addition, an online landing page has been created at NOAA Paleoclimatology

(https://www.ncdc.noaa.gov/paleo/study/22732) where all datasets are available as LiPD files and Matlab and R serializations.

(2) Output time series: The 100-year binned raw and standardized data for each proxy record, and the regional averages are available at the NOAA Paleoclimatology URL listed above. These enable future users to test different standardization approaches and to compare our composites with subsequent studies.

**Author contributions:**

All of the authors contributed to data compilation, study design, data interpretation, and manuscript preparation. NM, DK, and BS coordinated the workshops that facilitated the project. CR and NM processed the dataset and, with BS, conducted the
10   final statistical analyses and generated figures and tables. BS led the manuscript writing.

**Competing Interests:**

The authors declare that they have no conflict of interest.

**Acknowledgements:** We thank the U.S. Geological Survey Powell Center for Analysis and synthesis, Past Global Changes
15   (PAGES), and the University of Wyoming Roy J. Shlemon Center for Quaternary Studies for supporting the working group meetings. NSF award 1602105 to DK, CR and NM supported the analyses. We also thank all data contributors and the participants of the Powell Center working group meetings that lead to the project, including J. Cole, A. Viau, J. Rodsill, D. Willard, K. Anchukaitis, S. St. George, E. Cook, F. Jaume-Santero, and H. Beltrami, as well as J. Marsicek. Any use of trade, firm, or product names is for descriptive purposes only and does not imply endorsement by the U.S. Government.

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

**Table 1.** Hydroclimate record data including latitude (Lat), longitude (Long), and elevation (Elev). Records sensitive to annual precipitation (P) and effective moisture (P-E) are indicated as are records that have been calibrated to provide quantitative reconstructions (Cal). Sign refers to whether the variable has a positive or negative relationship to the hydroclimate inference (e.g., to effective moisture).

| Site name | Political unit | Lat (°) | Long (°) | Elev (m) | Archive | Variable | Sign | P | P-E | Cal | Reference | Original data URL |
|---|---|---|---|---|---|---|---|---|---|---|---|---|
| Abbott Lake | California | 36.23 | -121.48 | 286 | lake | particle size | + | | | | Hiner et al., 2016) | https://www.ncdc.noaa.gov/paleo/st |
| Basin Pond | Maine | 44.46 | -70.05 | 132 | lake | pollen | + | X | | X | Gajewski (1988) | https://www.ncdc.noaa.gov/paleo/st |
| Bat Cave | New Mexico | 32.2 | -104.4 | 1300 | speleothem | $\delta^{13}C$ | - | | X | | Asmerom et al. (2013) | URL pending from NOAA-paleo |
| Beaver Lake | Nebraska | 42.46 | -100.67 | 905 | lake | diatom | + | | X | | Schmieder et al. (2011) | URL pending from NOAA-paleo |
| Berry Pond | Massachusetts | 42.51 | -73.32 | 631 | lake | pollen | + | X | | X | Marsicek et al. (2013) | URL pending from NOAA-paleo |
| Big Lake | British Columbia | 51.67 | -121.45 | 1030 | lake | diatom | + | | X | | Cumming et al. (2002) | URL pending from NOAA-paleo |
| Bison Lake | Colorado | 39.76 | -107.35 | 3255 | lake | $\delta^{18}O$ | - | | | | Anderson (2011) | https://www.ncdc.noaa.gov/paleo/st |
| Blood Pond | Massachusetts | 42.08 | -71.96 | 214 | lake | pollen | + | X | | X | Marsicek et al. (2013) | https://www.ncdc.noaa.gov/paleo/st |
| Buckeye Creek Cave | West Virginia | 37.98 | -80.4 | 600 | speleothem | Sr/Ca | - | | | | Springer et al. (2008) | https://www.ncdc.noaa.gov/paleo/st |
| Bufflehead Pond | Minnesota | 44.99 | -93.54 | 515 | lake | stratigraphy | - | | X | | Shuman et al. (2009b) | URL pending from NOAA-paleo |
| Castor Lake | Washington | 48.54 | -119.56 | 594 | lake | reflectance | - | | X | | Nelson et al. (2011) | https://www.ncdc.noaa.gov/paleo/st |
| Castor Lake | Washington | 48.54 | -119.56 | 594 | lake | $\delta^{18}O$ | - | | X | | Steinman et al. (2012) | https://www.ncdc.noaa.gov/paleo/st |
| Chauvin Lake | Alberta | 52.68 | -110.1 | 625 | lake | diatom | - | | X | | Laird et al. (2003) | URL pending from NOAA-paleo |
| Chilibrillo Cave | Panama | 9.2 | -79.7 | 60 | speleothem | $\delta^{18}O$ | - | | X | | Lachniet et al. (2004) | https://www.ncdc.noaa.gov/paleo/st |
| Clear Pond | New York | 43.74 | -74.2 | 587 | lake | pollen | + | X | | X | Gajewski (1988) | URL pending from NOAA-paleo |
| Coldwater Lake | North Dakota | 46.01 | -99.08 | 594 | lake | diatom | - | | X | | Fritz et al. (2000) | URL pending from NOAA-paleo |
| Conroy Lake | Maine | 46.29 | -67.88 | 136 | lake | pollen | + | X | | X | Gajewski (1988) | https://www.ncdc.noaa.gov/paleo/st |
| Crevice Lake | Montana | 45 | -110.58 | 1713 | lake | $\delta^{18}O$ | - | | X | | Stevens and Dean (2008) | URL pending from NOAA-paleo |
| Davis Pond | Massachusetts | 42.14 | -73.41 | 214 | lake | stratigraphy | - | | X | X | Newby et al. (2011) | https://www.ncdc.noaa.gov/paleo/st |
| Deep Pond | Massachusetts | 41.56 | -70.64 | 23 | lake | pollen | + | X | | X | Marsicek et al. (2013) | https://www.ncdc.noaa.gov/paleo/st |
| Deep Pond | Massachusetts | 41.56 | -70.64 | 23 | lake | stratigraphy | - | | X | X | Marsicek et al. (2013) | https://www.ncdc.noaa.gov/paleo/st |
| Dixie Lake | Ontario | 49.83 | -93.95 | 398 | lake | diatom | + | | X | | Laird et al. (2012) | URL pending from NOAA-paleo |
| Dune Lake | Alaska | 64.42 | -149.9 | 134 | lake | $\delta^{13}C$ | - | | X | | Finney et al. (2012) | https://www.ncdc.noaa.gov/paleo/st |
| DYE3 | Greenland | 65.18 | -43.83 | 2479 | ice | accumulation | + | X | | X | Andersen et al. (2006) | http://www.iceandclimate.nbi.ku.dk 2006_Annual_Accumulation_22Ma |
| East Lake | Nunavut | 74.53 | -109.32 | 5 | lake | varve | + | | | | Cuven et al. (2011) | https://www.ncdc.noaa.gov/paleo/st |
| ELA Lake 239 | Ontario | 49.67 | -93.73 | 386 | lake | diatom | + | | X | | Laird et al. (2012) | URL pending from NOAA-paleo |

| Site | Region | Lat | Lon | Elev | Type | Proxy | +/- | | | Reference | URL |
|---|---|---|---|---|---|---|---|---|---|---|---|
| ELA Lake 442 | Ontario | 49.77 | -93.82 | 411 | lake | diatom | + | X | | Laird et al. (2012) | URL pending from NOAA-paleo |
| Elk Lake | Minnesota | 47.11 | -95.13 | 457 | lake | geochemical | - | X | | Dean et al. (1994) | https://www.ncdc.noaa.gov/paleo/st |
| Emerald Lake | Colorado | 39.15 | -106.41 | 3053 | lake | stratigraphy | - | X | | Shuman et al. (2014) | URL pending from NOAA-paleo |
| Fish Lake | Colorado | 37.25 | -106.68 | 3718 | lake | composite | - | | | Routson et al. (2016b) | https://doi.org/10.1371/journal.pone |
| Foy Lake | Montana | 48.17 | -114.35 | 1006 | lake | $\delta^{18}O$ | - | X | | Stevens et al. (2006) | URL pending from NOAA-paleo |
| Fresh Pond | Rhode Island | 41.16 | -71.58 | 28 | lake | pollen | + | X | X | Marsicek et al. (2013) | URL pending from NOAA-paleo |
| Gall Lake | Ontario | 50.23 | -91.45 | 365 | lake | diatoms | + | X | | Laird et al. (2012) | URL pending from NOAA-paleo |
| GISP2 | Greenland | 72.6 | -38.5 | 3209 | ice | accumulation | + | X | X | Meese et al. (1994) | https://www.ncdc.noaa.gov/paleo/st |
| GRIP | Greenland | 72.58 | -37.64 | 3237 | ice | accumulation | + | X | X | Andersen et al. (2006) | http://www.iceandclimate.nbi.ku.dk 2006_Annual_Accumulation_22Ma |
| Hells Kitchen Lake | Wisconsin | 46.19 | -89.7 | 502 | lake | pollen | + | X | X | Gajewski (1988) | https://www.ncdc.noaa.gov/paleo/st |
| Hidden Lake | Colorado | 40.51 | -106.61 | 2708 | lake | stratigraphy | - | X | | Shuman et al. (2009a) | URL pending from NOAA-paleo |
| Humboldt Lake | Saskatchewan | 52.13 | -105.1 | 552 | lake | diatom | - | X | | Laird et al. (2003) | URL pending from NOAA-paleo |
| Irwin Smith Bog | Michigan | 45.03 | -83.62 | 223 | bog | testate amoeba | - | X | | Booth et al. (2012) | https://doi.org/10.6084/m9.figshare. |
| Jenning Cave | Florida | 29.2 | -82.2 | 30 | cave | lipid $\delta^{13}C$ | - | | | Polk et al. (2013) | https://doi.org/10.1016/j.chemgeo.2 |
| Jones Lake | Montana | 47.05 | -113.14 | 1248 | lake | $\delta^{18}O$ | - | X | | Shapely et al. 2009 | URL pending from NOAA-paleo |
| Juxtlahuaca Cave | Guerrero | 17.4 | -99.2 | 934 | speleothem | $\delta^{18}O$ | - | | | Lachniet et al. (2012) | https://www.ncdc.noaa.gov/paleo/st |
| Lac le Caron | Quebec | 52.28 | -75.83 | 248 | bog | testate amoeba | + | X | | Loisel and Garneau (2010) | URL pending from NOAA-paleo |
| Lago El Gancho | Nicaragua | 11.9 | -85.92 | 44 | lake | $\delta^{18}O$ | - | X | | Stansell et al. (2012) | https://www.ncdc.noaa.gov/paleo/st |
| Laguna de Felipe | Dominican Republic | 18.8 | -70.88 | 1005 | lake | $\delta^{18}O$ | - | X | | Lane et al. (2009) | URL pending from NOAA-paleo |
| Lake Chichancanab | Quintana Roo | 19.83 | -88.75 | 15 | lake | sulphur | - | X | | Hodell (1995) | https://www.ncdc.noaa.gov/paleo/st |
| Lake Elsinore | California | 33.67 | -117.35 | 379 | lake | particle size | + | | | Kirby et al. (2010) | https://www.ncdc.noaa.gov/paleo/st |
| Lake Miragoane | Nippes | 18.4 | -73.05 | 14 | lake | $\delta^{18}O$ | - | X | | Hodell et al. (1991) | URL pending from NOAA-paleo |
| Lake of the Woods | Wyoming | 43.48 | -109.89 | 2820 | lake | stratigraphy | - | X | X | Pribyl and Shuman (2014) | https://www.ncdc.noaa.gov/paleo/st |
| Lake Peten-Itza | Peten | 16.92 | -89.83 | 110 | lake | $\delta^{18}O$ | - | X | | Curtis et al. (1998) | https://www.ncdc.noaa.gov/paleo/st |
| Lake Punta Laguna | Yucatan | 20.63 | -87.62 | 14 | lake | $\delta^{18}O$ | - | X | | Curtis et al. (1996) | https://www.ncdc.noaa.gov/paleo/st |
| Lake Winnipeg | Manitoba | 50.57 | -96.83 | 217 | lake | $\delta^{18}O$ | - | X | | Buhay et al. (2009) | URL pending from NOAA-paleo |
| Lime Lake | Washington | 48.87 | -117.34 | 780 | lake | $\delta^{18}O$ | + | X | X | Steinman et al. (2012) | https://www.ncdc.noaa.gov/paleo/st |
| Little Pond Royalston | Massachusetts | 42.68 | -72.19 | 302 | lake | pollen | + | X | X | Marsicek et al. (2013) | URL pending from NOAA-paleo |
| Little Raleigh | Ontario | 49.45 | -91.89 | 457 | lake | diatom | + | X | | Laird et al. (2012) | URL pending from NOAA-paleo |

| Name | Location | Lat | Lon | Elev | Type | Proxy | +/- | | | | Reference | URL |
|---|---|---|---|---|---|---|---|---|---|---|---|---|
| Little Windy Hill | Wyoming | 43.48 | -109.89 | 2821 | lake | stratigraphy | - | | X | X | Pribyl and Shuman (2014) | https://www.ncdc.noaa.gov/paleo/st |
| Lower Bear Lake | California | 34.25 | -116.91 | 2059 | lake | C/N | + | | | | Kirby et al. (2012) | https://www.ncdc.noaa.gov/paleo/st |
| Marcella Lake | Yukon Territory | 60.07 | -133.81 | 749 | lake | $\delta^{18}O$ | - | | | | Anderson et al. (2007) | https://www.ncdc.noaa.gov/paleo/st |
| MB01 | Nunavut | 69.81 | -112.08 | 290 | lake | pollen | + | X | | X | Peros and Gajewski (2008) | https://www.ncdc.noaa.gov/paleo/st |
| Meekin Lake | Ontario | 49.82 | -94.77 | 353 | lake | diatom | + | | X | | Laird et al. (2012) | URL pending from NOAA-paleo |
| Minden Bog | Michigan | 43.61 | -82.83 | 240 | bog | testate amoeba | - | | X | | Booth et al. (2012) | URL pending from NOAA-paleo |
| Minnetonka Cave | Idaho | 42.09 | -111.52 | 2347 | speleothem | $\delta^{13}C$ | + | | | | Lundeen et al. (2013) | URL pending from NOAA-paleo |
| Moon Lake (ND | North Dakota | 46.32 | -98.16 | 444 | lake | diatom | - | | X | | Laird et al. (1996) | https://www.ncdc.noaa.gov/paleo/st |
| N14 | Greenland | 59.98 | -44.18 | 101 | lake | BSi | + | | | | Andresen et al. (2004) | URL pending from NOAA-paleo |
| New Long Pond | Massachusetts | 41.85 | -70.68 | 29 | lake | stratigraphy | - | | X | X | Newby et al. (2009) | URL pending from NOAA-paleo |
| NGRIP | Greenland | 75.1 | -42.32 | 2922 | ice | accumulation | + | X | | X | Andersen et al. (2006) | http://www.iceandclimate.nbi.ku.dk 2006_Annual_Accumulation_22Ma |
| No Bottom Lake | Nantucket | 41.29 | -70.11 | 8 | lake | pollen | + | X | | X | Marsicek et al. (2013) | URL pending from NOAA-paleo |
| Nora Lake | Manitoba | 50.47 | -99.94 | 638 | lake | diatom | - | | X | | Laird et al. (2003) | URL pending from NOAA-paleo |
| North Pond (Athabasca) | Alberta | 58.8 | -110.71 | 213 | lake | C/N | + | | | | Wolfe et al. (2011) | URL pending from NOAA-paleo |
| Ongoke Lake | Alaska | 59.25 | -159.42 | 75 | lake | diatom | + | | X | | Chipman et al. (2008) | https://www.ncdc.noaa.gov/paleo/st |
| Oregon Caves | Oregon | 42.08 | -123.42 | 1390 | speleothem | $\delta^{13}C$ | + | | | | Ersek et al. (2012) | https://www.ncdc.noaa.gov/paleo/st |
| Oro Lake | Saskatchewan | 49.78 | -105.33 | 686 | lake | diatom | - | | X | | Michels et al. (2007) | URL pending from NOAA-paleo |
| Path Lake | Nova Scotia | 43.87 | -64.93 | 15 | lake | pollen | + | X | | X | Neil et al. (2014) | https://www.ncdc.noaa.gov/paleo/st |
| Pinhook Lake | Indiana | 41.61 | -86.85 | 259 | bog | testate amoeba | - | | X | | Booth et al. (2012) | URL pending from NOAA-paleo |
| Pyramid Lake | California | 40.02 | -119.56 | 1157 | lake | $\delta^{18}O$ | - | | X | | Benson et al. (2002) | https://www.ncdc.noaa.gov/paleo/st |
| Renner Lake | Washington | 48.78 | -118.19 | 754 | lake | $\delta^{18}O$ | - | | X | | Steinman et al. (2012) | https://www.ncdc.noaa.gov/paleo/st |
| Rice Lake | North Dakota | 48.01 | -101.53 | 620 | lake | Mg/Ca | - | | X | | Yu and Ito (1999) | https://www.ncdc.noaa.gov/paleo/st |
| Rogers Lake | Connecticut | 41.21 | -72.17 | 12 | lake | pollen | + | X | | X | Marsicek et al. (2013) | URL pending from NOAA-paleo |
| Round Lake | Nebraska | 48.42 | -101.5 | 1064 | lake | diatom | + | | X | | Schmieder et al. (2011) | URL pending from NOAA-paleo |
| Saco Bog | Maine | 43.55 | -70.46 | 45 | bog | testate amoeba | - | | X | | Clifford and Booth (2013) | URL pending from NOAA-paleo |
| Sidney Bog | Maine | 44.39 | -69.78 | 91 | bog | testate amoeba | - | | X | | Clifford and Booth (2013) | URL pending from NOAA-paleo |
| South Rhody Bog | Michigan | 46.56 | -86.07 | 290 | bog | testate amoeba | - | | X | | Booth et al. (2012) | URL pending from NOAA-paleo |
| SS1381 | Greenland | 67.01 | -51.1 | 196 | lake | mineral | + | | X | | Anderson et al. (2012) | https://www.ncdc.noaa.gov/paleo/st |

| | | | | | | | | | | |
|---|---|---|---|---|---|---|---|---|---|---|
| SS16 | Greenland | 66.91 | -50.46 | 477 | lake | diatom | + | X | | Perren et al. (2012) | https://www.ncdc.noaa.gov/paleo/st |
| Steel Lake | Minnesota | 46.97 | -94.68 | 415 | lake | $\delta^{18}O$ | - | X | | Tian et al. (2006) | https://www.ncdc.noaa.gov/paleo/st |
| Swan Lake | Nebraska | 42.16 | -99.03 | 702 | lake | diatom | + | X | | Schmieder et al. (2011) | URL pending from NOAA-paleo |
| Takahula Lake | Alaska | 67.35 | -153.67 | 275 | lake | $\delta^{18}O$ | - | X | | Clegg and Hu (2010) | https://www.ncdc.noaa.gov/paleo/st |
| Tzabnah Cave | Yucatan | 20.75 | -89.47 | 20 | speleothem | $\delta^{18}O$ | - | X | | Medina-Elizalde et al. (2010) | https://www.ncdc.noaa.gov/paleo/st |
| Victoria Island | Nunavut | 69.8 | -112.06 | 290 | lake | pollen | + | X | X | Peros and Gajewski (2008) | https://www.ncdc.noaa.gov/paleo/st |
| Wolverine Lake | Alaska | 67.1 | -158.91 | 85 | lake | accumulation | - | X | | Mann et al. (2002) | URL pending from NOAA-paleo |
| Yellow Lake | Colorado | 39.65 | -107.35 | 3170 | lake | $\delta^{18}O$ | - | | | Anderson (2012) | https://www.ncdc.noaa.gov/paleo/st |
| Yok Balum Cave | Toledo | 16.21 | -89.07 | 366 | speleothem | $\delta^{18}O$ | - | X | | Kennett et al. (2012) | https://www.ncdc.noaa.gov/paleo/st |
| Zaca Lake | California | 34.78 | -120.04 | 737 | lake | particle size | + | | | Kirby et al. (2014) | https://www.ncdc.noaa.gov/paleo/st |

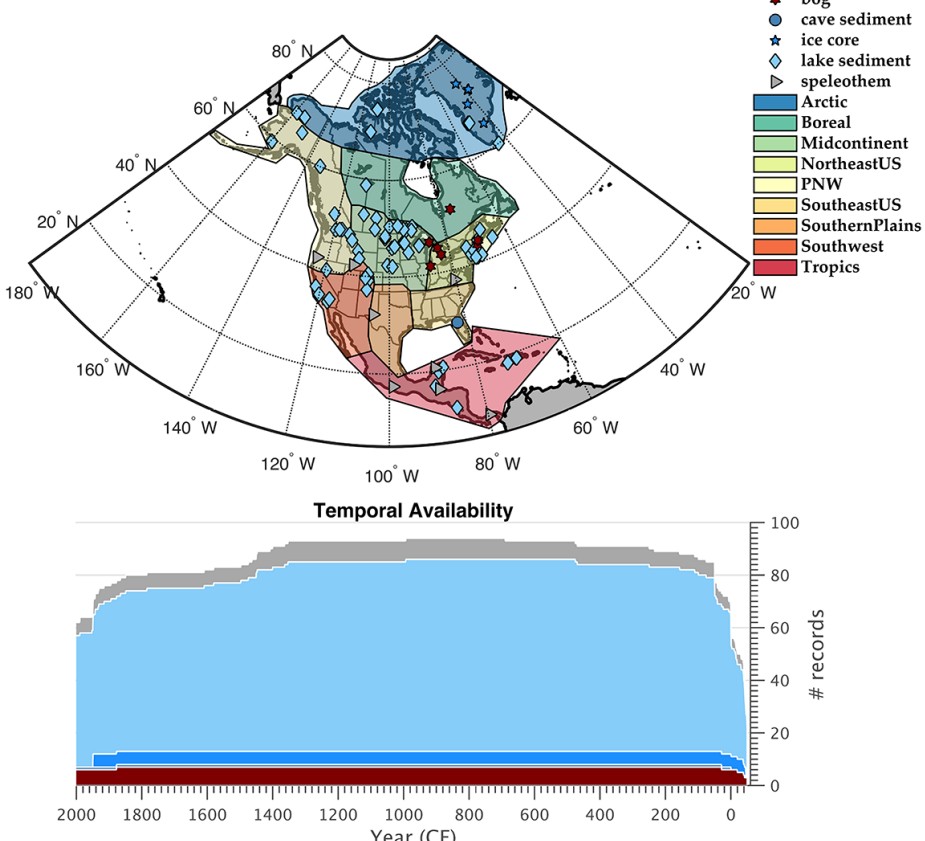

**Figure 1: Temporal and spatial distribution of the North American paleo-hydroclimate records used in this study. Top: geographical distribution by archive type indicated by symbol colors and shapes. The differently colored areas on the map delimit the regions used for principal components analyses (e.g., PNW, Pacific Northwest). Bottom: The temporal distribution of records in the dataset. Colors correspond to the archive types in the map.**

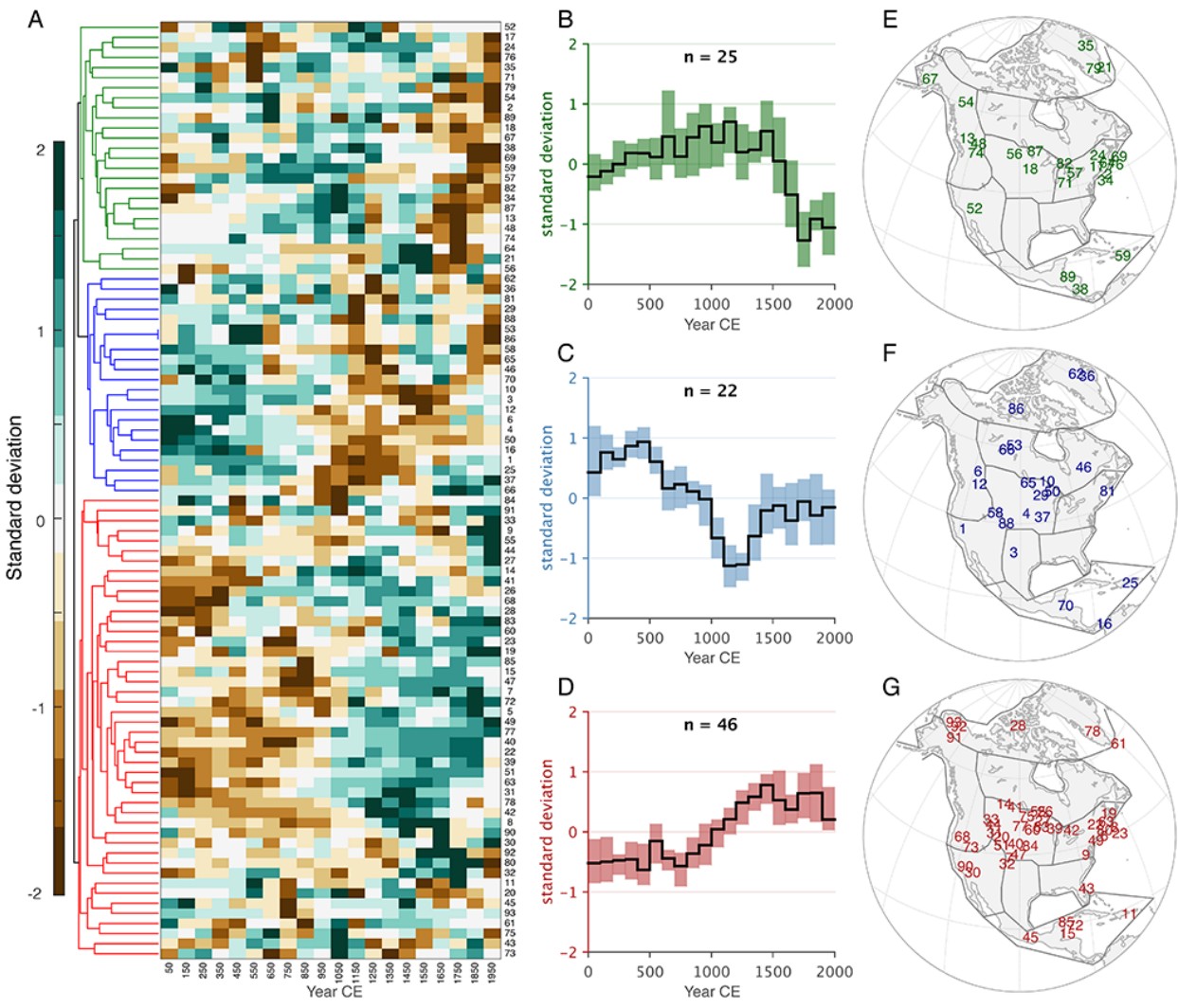

**Figure 2: A) Heat map, cluster analysis, and dendrogram of the standardized hydroclimate records used in this study. The clustering shows the dominant groups of temporal patterns represented by the hydroclimate records. B-D) Average composites of the respective colored clusters in panel A. The shading shows the 95% bootstrap confidence intervals with 500 iterations. E-F) The spatial distribution of records in each cluster; numbers correspond with those in panel A.**

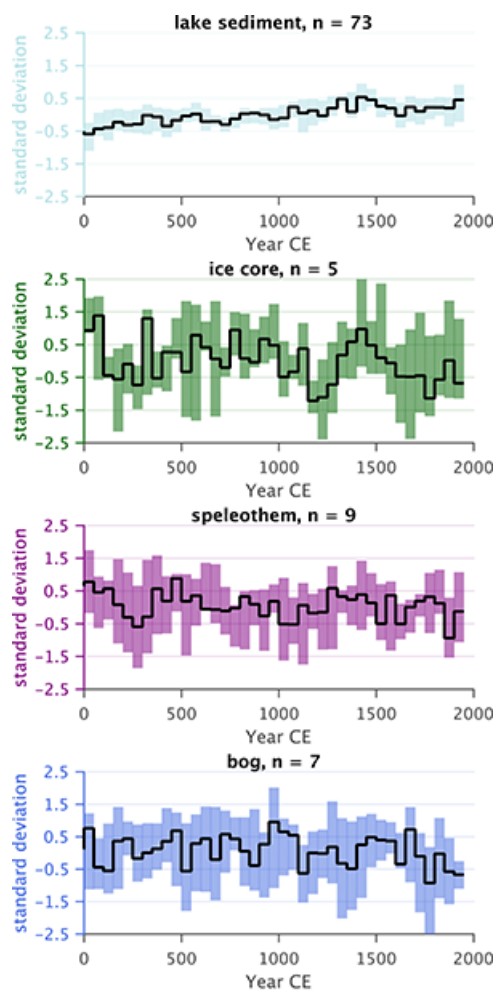

Figure 3: Mean z-score time series for each archive type that contains >1 record.

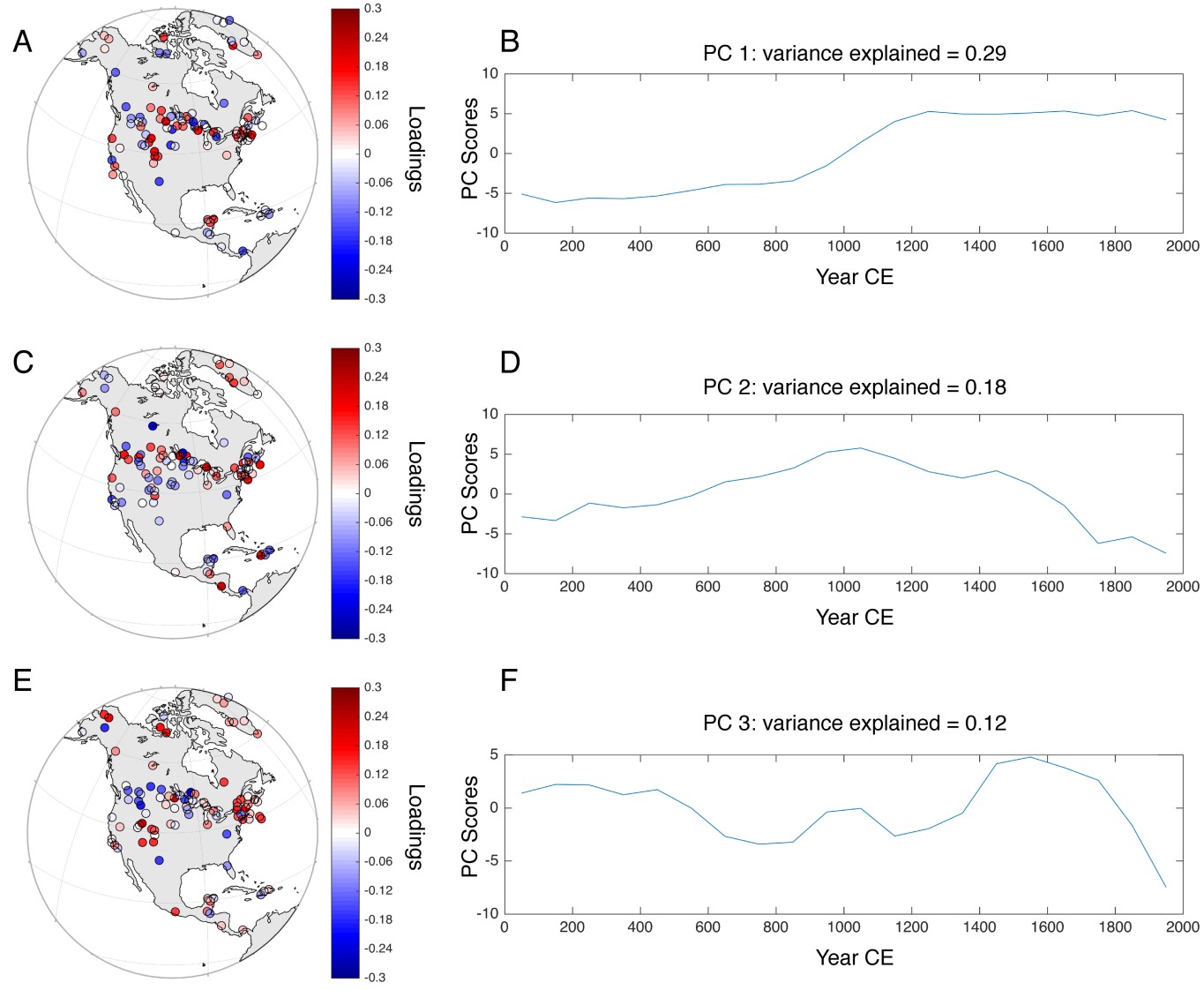

**Figure 4: Results of empirical orthogonal function analysis (EOF) of the North American hydroclimate dataset showing (left) loadings associated with the first EOF, and (right) the average time series for each EOF.**

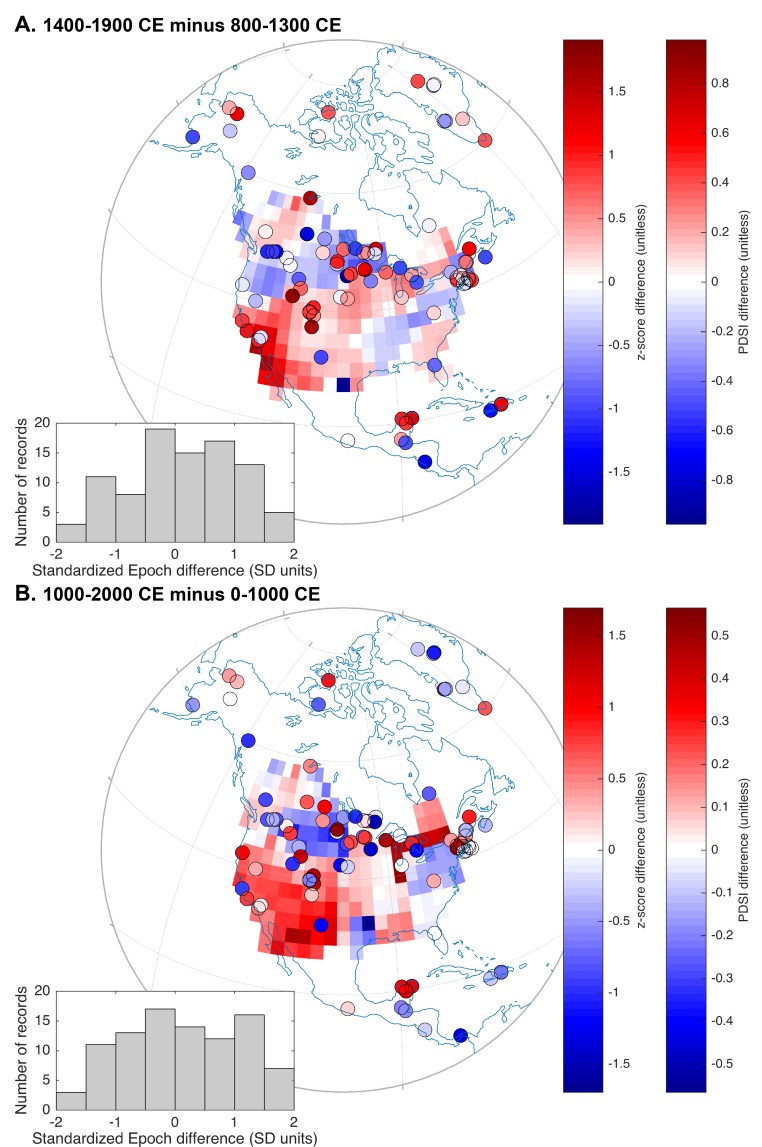

**Figure 5: Epoch differences showing A) the average of the Little Ice Age (1400-1900 CE) minus the average of the medieval climate anomaly (800-1300 CE) , and B) the average standardized value (SD) of the second millennium (1000-2000 CE) minus the average of the first millennium (0-1000 CE). Red symbols indicate that the older interval (epoch) was drier than the younger interval; blue symbols indicate the opposite. Circles show the datasets used here for comparison with the gridded North America Drought Atlas Palmer Drought Severity Index (PDSI) differences (Cook and Krusic, 2008); the right scale bar applies to the PDSI data. Histograms below show the total distribution of differences for each epoch from the dataset presented in this paper; both are skewed positively toward the older interval being drier than the younger interval.**

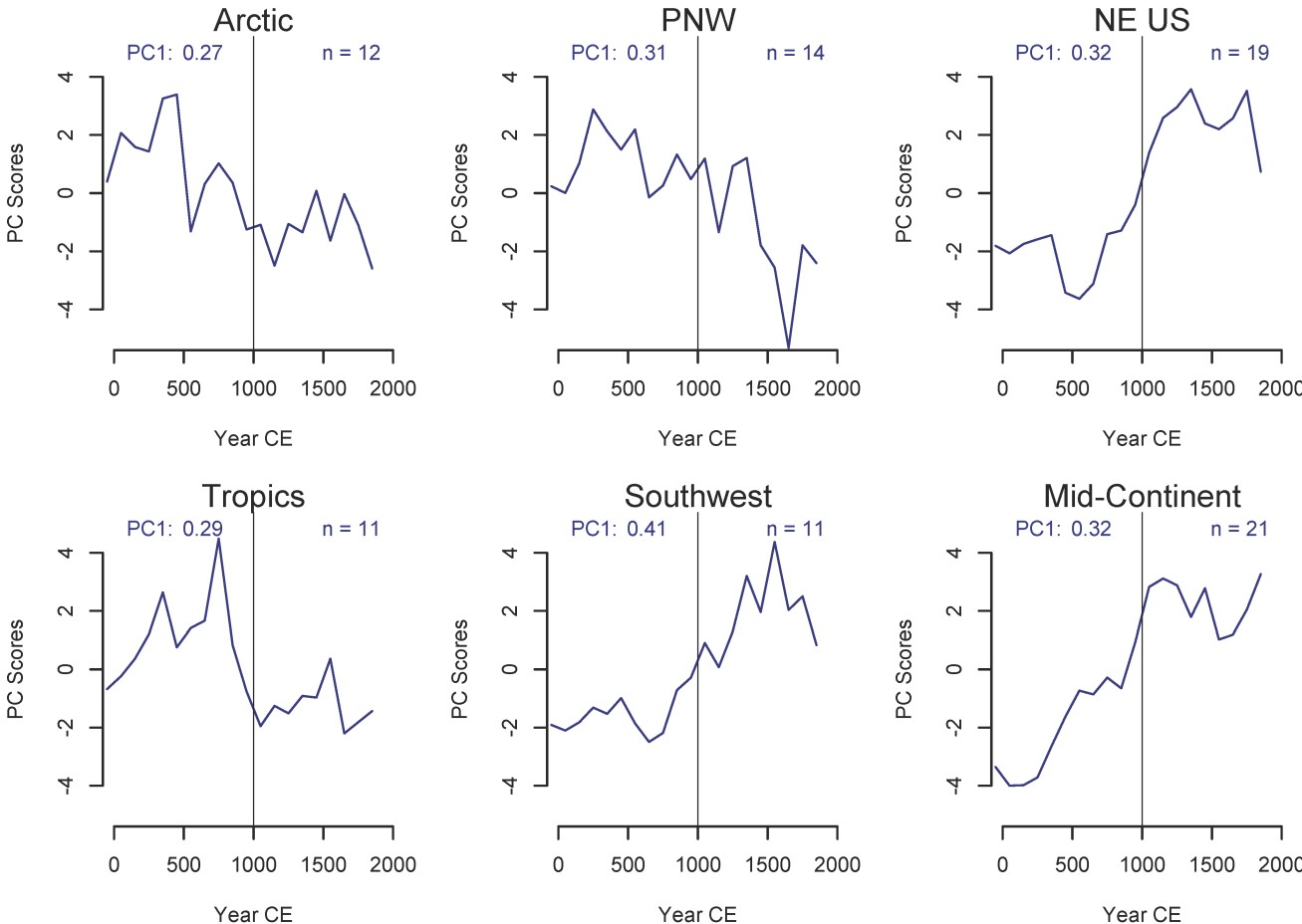

**Figure 6: Regional principal components analysis (PCA) scores showing the dominant modes of variability in six North American hydroclimate regions delineated in Figure 1. Numbers on the left represent the fraction of the variance explained by PC1 in each region. Vertical lines mark 1000 CE.**

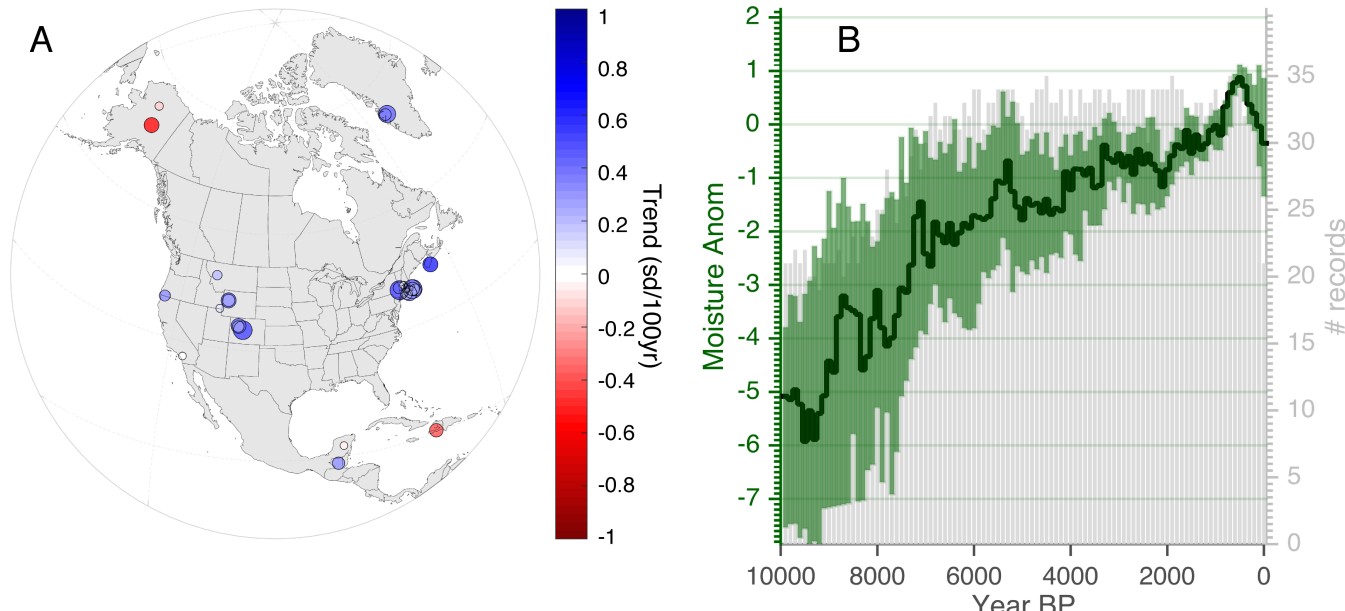

**Figure 7: Holocene context including records that extend 6000 years or longer in the North American hydroclimate dataset. A: Maps of Holocene-length trends. Blue colors show wetting trends, red colors show drying trends and the size of the symbol reflects the magnitude of the trend. B: The median Holocene composite of the z-scores of the Common Era records (Fig. 1), with 95% bootstrapped confidence intervals. Units represent standard deviations from the mean of the Common Era. Grey bars show the number of available records through time.**

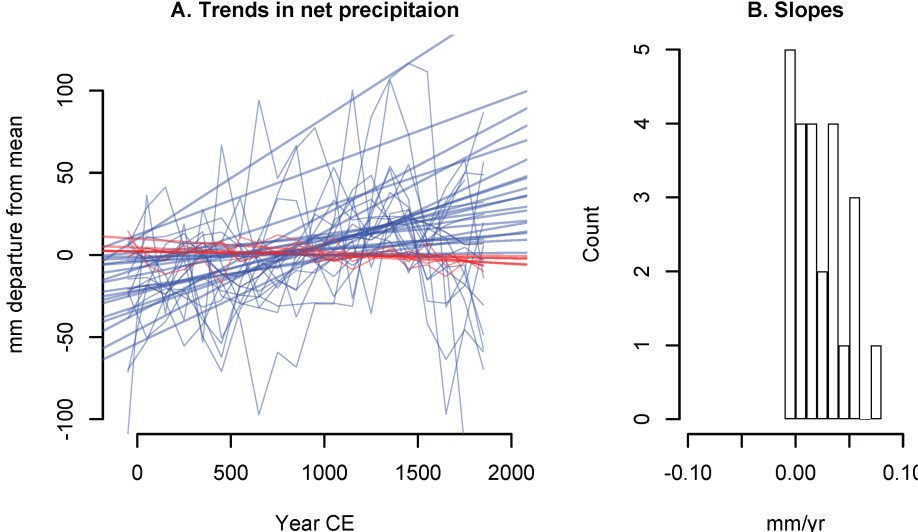

**Figure 8:** Time series and trends of all quantitative records presented as mm departures from their mean over the Common Era. Net precipitation is used here to include pollen-inferred annual precipitation, precipitation minus evapotranspiration changes estimated from lake levels, and ice core accumulation rates (calibrated records in Table 1). Blue time series indicate those with positive linear trends from 1-1500 CE; red time series have negative linear trends. Trend lines in A were fit using generalized linear models with a first-order autoregressive term. The slopes of the models appear as a histogram in B.