# Peer review of "Placing the Common Era in a Holocene Context: Millennial-tocentennial patterns and trends in the hydroclimate of North America over the past 2000 years"

_Climate of the Past, 2017_

## Short Comment (SC1) · 7 May 2017

The PAGES Data Stewardship Integrative Activity seeks to advance best practices for sharing data generated and assembled as part of all PAGES-related activities. As part of this activity, a team of reviewers has been constituted for the "Climate of the Past 2000 years" Special Issue. The data team is reviewing the data handling within each of the CP-Discussion papers in relation to the CP data policy and current best practices. The team has identified essential and recommended additions for each paper, with the goal of achieving a high and consistent level of data stewardship across the 2k Special

Issue. We recognize that an additional effort will likely be required to meet the high level of data stewardship envisaged, and we appreciate dedication and contribution of the authors. This includes the use of Data Citations (see example in supplement). We ask authors to respond to our comments as part of the regular open interactive discussion. If you have any questions about PAGES Data Stewardship principles, please contact any of us directly.

Best wishes for the success of your paper,

2k Special Issue Data Review Team (Darrell Kaufman, Nerilie Abram, Belen Martrat, Raphael Neukom, Scott St. George) and ex-officio team members (Marie-France Loutre, Lucien von Gunten)

Essential additions for this paper:

(1) As indicated in the current 'Data availability" section, the authors commit to archiving upon publication both (a) the input data listed in Table 1 and (b) the standardized hydrological time series and regional averages illustrated in Figure 2.

Please also note the supplement to this comment.

Please also note the supplement to this comment:
http://www.clim-past-discuss.net/cp-2017-35/cp-2017-35-SC1-supplement.pdf

---

## Short Comment (SC2) · 7 May 2017

Because I read through this article as part of the PAGES Data Review process, I thought I would also weigh in on another issue that caught my attention.

In the abstract and the Introduction, the authors note that tree-ring records are known to exhibit an upper-limit to their frequency response when used as a hydroclimatic proxy. Due to that limitation, tree-ring reconstructions of past drought often exhibit less low-frequency variance than other lower-resolution proxies at decadal or centennial timescales.

I don't take issue with you pointing out that tree rings have known limitations as hydroclimatic proxies. But given that you're arguing that proxies from lakes, caves, bogs and ice can provide useful information about longer-term changes, I think it's also necessary to acknowledge the limitations of those archives. As you point out in Section 2.1.2, many of the proxies included in this compilation reflect aspects of hydrology - lake level, water depth, salinity, run-off - that are strongly influenced by hydrological storage. And because storage imparts long-term memory, it's reasonable to assume that these proxies will exhibit more variance at low frequencies than a proxy like tree rings, which is going to be more closely tuned to precipitation or soil moisture.

Kathleen Huybers and colleagues published a nice paper in Climate Dynamics last year http://link.springer.com/article/10.1007/s00382-015-2798-4) showing that, even if a watershed is forced with random climate, because lakes integrate those fluctuations over time, interannual climate variations by themselves can cause substantial decadal or multidecadal swings in lake level. Obviously the relative importance of storage is going to vary across this large and diverse dataset. But if you're going to point out the limits of one proxy, it's only fair that you consider the potential issues or confounding effects inherent to the records that are included in this compilation.

---

## Referee Comment (RC1) · Anonymous Referee #1 · 22 May 2017

The paper presents an interesting new synthesis of a large paleo hydroclimate dataset for North America during the past 2kyr. The objective is to identify the spatiotemporal patterns and magnitude of changes in centennial and millennial-scale variations in hydroclimate. Contrasts are drawn between hydrologic changes in the MCA versus the LIA, and between the first and second millennia. A variety of exploratory analyses are conducted using cluster analysis, EOF/PCA, and GLMs. Results show a wetting trend at most sites, but some regions showed a drying trend. Differences exist between the MCA and LIA, but are largely part of longer-term trends. Overall, there is very high spatial variability across the sites, and almost no regional coherence, but distinct temporal trends emerge from the data.

[Figure]

The paper is clear and the findings consistent with the data, but the present design raises some questions about the approach. The approach is generally inductive and bottom-up, but a primary component of the analysis is focused on regions, which is not well justified. Regions are imposed on the data without explanation about how they were determined. And in fact, the data and analyses conducted seem inconsistent with the regional approach given that the results from the cluster, EOF, or epoch analyses do not yield clearly identifiable or coherent spatial patterns. As a result, the PCA-by-region analysis does not seem appropriate, as the trends in different regions are very similar in several cases (e.g. NE US and Mid-continent, or the Arctic and PNW). There is also no discussion connecting the paleo data to modern spatiotemporal patterns in hydroclimate. As a result, the paper lacks context, raising questions about how the spatial patterns in the reconstructions compare with current observations or historical variations in precipitation or moisture availability.

Given that the study includes many calibrated records (27%), the analyses also raise the question of whether this subset of records alone might provide a more coherent representation of change than the broader dataset. It would be helpful to resolve this in order to help motivate (or constrain) future research. A related question is whether there were differences in the annual precipitation reconstructions based on the fossil pollen records versus those used to reconstruct changes in effective moisture. It seems plausible given that one could expect quite different patterns to emerge from these two types of reconstructions depending on temperature variations.

Last, although the stated emphasis of the synthesis is on records that primarily capture multi-centennial trends, the decision to include ice accumulation but not tree-rings seems somewhat arbitrary, especially when the objective is to characterize hydroclimate from N America, while omitting one of (if not the) most important data sources of such information from the analysis, especially when it is publicly available. Thus it is not possible given the current study design to know how variations in these results compare with those in drought characteristics (e.g. location, severity, frequency, duration),

a key component of hydroclimate.

Specific comments:

Pg 3 line 16: "Most of the record have been calibrated to a specific hydrologic variable..." seems to contradict the prior statement on line 1 "Twenty-five records (27%) provide calibrated reconstructions of hydroclimatic variables)...". Although there is a contrast between hydrologic and hydroclimatic, the units (e.g. millimeters of precipitation versus "annual precipitation") are the same and thus the terminology and differences could use some clarification.

Pg 5 line 25: How does SSA estimate missing values? Please add some detail given the importance to the methods and the fact that this is a common problem in paleo studies.

Pg 6 line 14: How were the regions defined? How was PCA conducted on regions that appear to have only one or two sites?

Pg 6 line 18: What does "hydrological interferences" mean? (inferences?)

Pg 7 line 2: Top dendrogram cluster is green (not blue), no? Maybe refer to clusters by their color, or label the clusters to avoid ambiguous refs (e.g. "in one of the clusters..." (pg. 6, line 29).

Pg 7, line 5: There does seem to be a clear trend in 2C. It is visible in the heat map also as shifts from wetter to drier (opposite from the red clusters/panel D) though weaker. Standardize the y-axes?

Pg 10, line 14: this one sentence refers to complex hydroclimate but leaves one wondering what patterns at all exist in the modern or historical data, and in need of more information about the complexity to evaluate or make sense of the paleo data. This paragraph also raises the question of whether the authors considered elevation or other physical drivers or spatial patterns.

[Figure]

Pg. 11-12 Section 4.3. This discussion section is lacking information about the regions. Only the PNW is discussed and several other regions are not even mentioned. If the regional approach is retained, more lit review and discussion is needed about each of the regions to give meaning to the new results and aid interpretation.

Pg 12, line 1-2, add info about what the temperature trend was (i.e. cooling) to clarify sentence.

Table 1 - Some information about the number and types of dates at least (and source materials, and error if available) would be helpful for understanding how much uncertainty is coming from dating of the records.

Figure 1 – the site map is not very legible as the symbol sizes are too large relative to the land. Also, why are nine regions marked but only six appear in the PCA (Fig. 5).

Figure 2 - indicate wet versus dry conditions on panels B-D. Also, for panel B-D, it would be helpful to standardize the axes so it easier to see weak/strong trends. For panels E-G, the lat/long grid is much less important than the site details – a rectangular bounding box would allow the land area to be enlarged to better show more of the site numbers, which are currently mostly illegible.

Figure 4 – perhaps show the distribution of values below each map to emphasize whether more sites are showing drier or wetter conditions. It is difficult to see any pattern in these maps, or to know whether any apparent clusters are meaningful.

―――――――――――――

---

## Referee Comment (RC2) · Anonymous Referee #2 · 17 Aug 2017

The authors have put considerable effort into collating and summarizing a large pool of proxy hydroclimate data from Central and North America using an impressive suite of statistical techniques. The focus was on centennial and longer trends over the Common Era (CE) with particular emphasis on contrasting the first and second millennium as well as the Medieval period (800-1300 CE) and Little Ice Age (1400-1900). The paper is clearly written and well structured with excellent figures. However, my concerns relate mainly to what has not been included and I have outlined these concerns in sections below.

a) Comparison to tree-ring based reconstructions (drought atlases). I was very sur-

[Figure]

prised about the lack of any direct comparison to published drought atlases derived from tree-rings (e.g. Cook et al. 2008 and Stahle et al. 2016). This is something that has also been raised by another reviewer. Clearly, the authors have the analytical skills to have addressed such comparisons. Their comment that tree-rings fail to preserve low frequency variance (lines 20-26, page 2) and therefore ignored, fails to recognise significant recent advances in tree-ring reconstructions (e.g. see "signal free" standardisation described in Melvin and Briffa 2008, 2014; and the methods applied to studies such as Stahle et al. 2016 and Cook et al. 2010, 2015). The lack of any direct comparison also ignores the fact that the drought atlases describe multidecadal to centennial length periods of drought and pluvials. The authors also discuss the possibility (Section 4.1, page 10) that some of their records may have also failed to capture long term trends due to detrending so this fact along with their inclusion of ice accumulation records makes the omission of the tree-rings seem arbitrary. My concern is, despite many approaches to analysing the data no clear spatially coherent patterns seem to emerge and I believe without the direct comparison to the tree-ring records the validity of the presented results remains hard to assess. I believe the paper needs to include comparisons and discussion about the published drought atlases given their clear geographical overlap.

b) Selection of windows of time. Several analyses involved the selection of time periods (or "windows of time") without any indication of the reasoning behind it. Let's start with Section 2.2.1. Why specifically 100-year bins? I know that a consistent window-length had to be chosen across all sites, but why was it 100? Given some sites dating uncertainties, how conservative is 100-years? This window length results in a maximum of 20 bins covering the last 2k. Was there a cohort of sites (say 25 or more) that had the potential for 50-year bins? In the opening introduction (line 29, page 2) the authors state "many have decadal resolution...". I would like to see the inclusion in Table 1 of information about the dating resolution and time-span covered by each site (and perhaps their autocorrelation). I would also like to recommend the inclusion in Table 1 of the Gini coefficient (Biondi and Qeadan, 2008) to help indicate those sites which show

strong records of past environmental variability. These additional parameters would then provide a basis that could be summarised (plotted?) and used as the rationale for the selected bin-window. Was there a missed opportunity here to present a smaller cohort but at higher temporal resolution or even split the database into two cohorts at different temporal resolutions? How many missing bins were there (an expanded Table 1 would help provide this information)? Why pick a 5-bin (500yr) moving window – how strong was the autocorrelation? The next aspect I think needs addressing is the selection of the CE time period. I understand the desire to present something aligned to the PAGES 2K initiatives but does this then stretch (i.e. require missing bins) the data of many sites or does it undervalue the temporal strength of many proxies? There is obviously a strong cohort of long proxies in the collection enabling the evaluation back through the Holocene (Section 3.5) so some discussion about the adoption of the CE period is I believe warranted. There is no discussion or comparison made about the nested approach used with the subset of the long cohort – e.g. what influence does the long cohort of sites have on the CE analyses (i.e. Figure 2 and 3)? I like the idea of epoch differencing between the LIA and MCA (despite the lack of any spatial coherence in Figure 4) but don't understand the rationale for the first versus second millennia comparison. The latter seems arbitrary, especially since it dissects the MCA window – a widely acknowledged period of climate importance. I would like to see epoch differencing over other windows. So for example, what about comparisons of the earlier period of 300-800 CE to the LIA and the MCA? We would then start to be able to appreciate how significant or different the LIA and/or MCA were. Is it possible to do epoch differencing of a moving 500-year period compared to the MCA (and the LIA)? I wonder also why a 500-year window was chosen for the MCA and LIA – how does the pattern change if a shorter window is chosen of say 400 years (900-1300 CE and 1400-1800 CE)? Is it possible to have a selected 500-year moving window that is compared to 5 randomly selected 100-year bins (like bootstrapping). If the millennia comparisons want to be retained, then perhaps do the same approach using only the long cohort of sites so that multiple millennia can be compared.

[Figure]

c) Selection of groupings Again, the selection of 9 geographic regions is presented without any explanation of the reasoning behind it. In Figure 1 the 9 different geographic regions are shown but only 6 of these are illustrated in Figure 5 presumably due to the lack of sites being located in the remaining 3 areas. I wonder then if some sort of biogeographic merging could be done so there aren't the gaps and to increase the associated sample depth (e.g. arctic plus boreal). Could a description of the characteristic climatic regime also be added about each of the regions as means of explaining why they have been used. As it stands, I agree with comments from the other referee about the PCA-by-region being somewhat inconsistent given the lack of regional coherence in the EOF results. The thorough statistical exploration of the database did not extend into looking at the specific proxy-types on their own, despite them being described in the methods (see Sections 2.1.2 - 2.1.5, pages 4 & 5). Could principal component analysis be done on the proxy types alone? This might help inform on the geographic patterns and / or the leading modes of variability in the whole dataset. I think the other referee also made a good suggestion of looking in more detail at the calibrated records.

References: Biondi F, Qeadan F (2008) Inequality in paleorecords. Ecology 89, 1056–1067. doi:10.1890/07-0783.1. Cook ER, Seager R, Kushnir Y, Briffa KR, Büntgen U, Frank D, Krusic PJ, Tegel W, van der Schrier G, Andreu-Hayles L, Baillie M, Baittinger C, Bleicher N, Bonde N, Brown D, Carrer M, Cooper R, Čufar K, Dittmar C, Esper J, Griggs C, Gunnarson B, Günther B, Gutierrez E, Haneca K, Helama S, Herzig F, Heussner K-U, Hofmann J, Janda P, Kontic R, Köse N, Kyncl T, Levanič T, Linderholm H, Manning S, Melvin TM, Miles D, Neuwirth B, Nicolussi K, Nola P, Panayotov M, Popa I, Rothe A, Seftigen K, Seim A, Svarva H, Svoboda M, Thun T, Timonen M, Touchan R, Trotsiuk V, Trouet V, Walder F, WaÅijny T, Wilson R, Zang C (2015) Old World megadroughts and pluvials during the Common Era. Science advances 1, e1500561. doi:10.1126/sciadv.1500561. Cook ER, Seager R, Heim RR, Vose RS, Herweijer C, Woodhouse C (2010) Megadroughts in North America: placing IPCC projections of hydroclimatic change in a long‐term palaeoclimate context. Journal of Quaternary

Science 25, 48–61. doi:10.1002/jqs.1303. Melvin TM, Briffa KR (2008) A 'signal-free' approach to dendroclimatic standardisation. Dendrochronologia 26, 71–86. doi:10.1016/j.dendro.2007.12.001. Melvin TM, Briffa KR (2014) CRUST: Software for the implementation of Regional Chronology Standardisation: Part 1. Signal-Free RCS. Dendrochronologia 32, 7–20. doi:10.1016/j.dendro.2013.06.002. Stahle DW, Cook ER, Burnette DJ, Villanueva J, Cerano J, Burns JN, Griffin D, Cook BI, Acuña R, Torbenson MCA, Szejner P, Howard IM (2016) The Mexican Drought Atlas: Tree-ring reconstructions of the soil moisture balance during the late pre-Hispanic, colonial, and modern eras. Quaternary Science Reviews 149, 34–60. doi:10.1016/j.quascirev.2016.06.018.

---

## Author Comment (AC1) · 15 Sep 2017

Thank you to the review team for their suggestions. We agree to lodge the specified datasets with a public repository, as indicated in the "data availability" section.

---

## Author Comment (AC2) · 15 Sep 2017

It is a good point that "low frequency" variation in systems with autoregressive memories of various durations can arise both from low frequency and high frequency climate forcing, and that it is not always clear which factor (or their interaction) is important for a given record.

We agree that all of the datasets involved have both strengths and limitations - and that they can have unique sensitivities to climate variations of different duration and in different variables. We propose to add brief discussions of limitations to sections 2.1.2-2.1.5 where we describe the different archives used in our analysis. We appreciate the

reference to the Huybers et al. paper.

Overall, we had not intended our comments or analysis as a critique of dendroclimate records, but we can see that our wording did not fully clarify our intention to evaluate the patterns in the types of data normally used to study long-term changes extending through the Holocene. Please see our related responses to the two reviews and our proposed title revision: "Placing the Common Era in a Holocene Context: Millennial-to-centennial patterns and trends in the hydroclimate of North America over the past 2000 years."

---

## Author Comment (AC3) · 15 Sep 2017

The reviewer provides several insightful points that we will use to improve the manuscript. As context, we would summarize our findings as evidence that long-term (millennial) hydroclimate trends affected North America during the Common Era, but that these trends were potentially small compared to both interannual variation and Holocene-length trends. The records and attendant spatial patterns are noisy because the signals of these trends and other centennial variations are small compared to the sensitivity of many archives used to study Holocene-scale variations. The points relate to three themes raised by the reviewer:

[Figure]

Regions and regional patterns: We agree that the spatial patterns do not consistently express strong spatial coherence. We, therefore, propose revisions to the text that explain the basis for the a priori regional assignments as well as additional discussion of the limitations of our dataset, which could produce noisy patterns (see also response to comment by S. St. George).

The regions were identified before our analysis as described by in a PAGES workshop report by McKay (2014, p. 100): "Based on the dominant airmasses, ecology, and the availability of proxy data throughout the continent, the group developed initial spatial targets for subcontinental. . . reconstructions." However, the distribution of data led to further divisions of the mid-continent and southern plains as well as the northeast and southeast US to ensure that geographically disparate locations were not averaged together; these decisions were guided by previously-published modern climatological pattern analyses (Mock, 1996) before analysis of the paleoclimate trends.

We propose to revise our text to more clearly explain the basis for the regions in section 2.1.1. We propose to insert the text along the lines of the following after the first sentence of section 2.1.1.:

"The nine regions were determined before we compiled the data considered here (McKay, 2014), and were based on the level 1 ecoregions of North America (Commission for Environmental Cooperation Working Group, 1997) and major patterns of covariance within modern climate data (Mock, 1996). Where spatial outliers existed separate from the main cluster of data within a region (e.g., data from Florida versus the northeast U.S.; northern versus southern Great Plains), we split our initial regions to ensure suitable representation of the data in our analysis. We used these designations to ask whether distinct trends were recognizable among commonly recognized regions and whether any trends have parallels to patterns of climate variation observed at finer time scales, such as north-south anti-phased moisture variability along the western margin of North America (Cayan, 1996; Wise and Dannenberg, 2014)."

In this way, we saw the analysis as related to modern climate patterns, although not as explicitly as raised by the reviewer. We propose to expand the Discussion section 4.1 to address potential parallels to historic patterns of variability within the noisy spatial patterns reconstructed by our analysis.

We understand the questions about the PCAs for each region, but would clarify that the analyses served to help us examine intra-regional coherency as well as inter-regional correlation. Figure 2 shows both the temporal and spatial dimensions of patterns regardless of our regional designations, whereas Figure 5 attempts to assess how much variance is shared (based on PCs) among records within each region. Therefore, in section 2.2.3, we proposed to add:

"The PCA-by-region analysis was conducted to evaluate the strength of any signals within each region, rather than simply calculating mean trends, and to assess potential correlations or shared signals across geographically distinct regions. The EOF analysis evaluates the latter from the perspective of the whole dataset."

Trends in the calibrated datasets: The reviewer raises an excellent point about the power of the calibrated datasets. We included Figure 7 as a means to demonstrate that the primary signal in these data is the dominant wetting trend. Unfortunately, these records are spatially clustered (primarily from the northeast U.S.) and may not be representative of the whole. Previous and forthcoming work has shown that no distinct differences exist between the pollen-inferred precipitation changes and P-E changes estimated from lake volume changes (Marlon et al., 2016; Marsicek et al., 2013). We propose to add these points to the text where suitable.

Criteria for data inclusion and overall goals: We had intended this manuscript to focus on the patterns observed in archives that are used to study Holocene-scale climate variations, and thus provide a bridge between the excellent work with dendroclimate records and studies of the whole interglacial. Therefore, we focused on archives that act as low-pass filters on paleoclimate changes. We agree that many different archives

from tree rings to lake sediments may retain low-frequency signals or that such signals could derive from the characteristics of high-frequency events, and that the inclusion of some annually layered records appears to create an inconsistent application of criteria regarding what data were examined.

We should clarify, however, that our primary goal was to determine the signals captured by the types of records used to study the Holocene. We admit that we did not make this aim clear, but propose changing our title to emphasize this aspect of the study (see also our response to Review #2). This goal inherently led us to include ice cores, especially because they represent nearly direct measurements of past precipitation. Tree ring records have been the dominant source of past hydroclimate information over the Common Era, and in many ways, our analysis was intended not as a critique of that excellent work, but rather as a complement intended to ask: what patterns of change are recorded by "everything else"?

Specific comments:

P. 3 Line 16 – We agree that we need to standardize the wording for clarity about hydroclimate versus hydrologic changes, and replace the word "calibrate" in this particular line with "interpreted to represent a specific climate variable."

P. 5 Line 25 – We will provide a sentence clarifying the application of SSA to missing values.

P. 6 Line 14 – As noted above, we propose to clarify about the regions. PCA was not applied where only one or two sites exist. Only the regions shown in Fig. 5 with >10 records were analyzed. We will add text to explain this point.

P. 6 Line 18 – "Interferences" was a typo and should be "inferences".

P. 7, Line 2 – We will rephrase to ensure that the references to the clusters in Figure 2 are clear. The suggestion to refer to them by color seems like it may be useful.

P. 7, Line 5 – We can see the trend that the reviewer mentions and will try to describe

it. However, the primary feature identified as a deviation for the mean in the EOF is the Medieval Anomaly. The EOF contrasts the early portion of these records with the Medieval Period, but we agree that a background trend also exists because the records never return fully to their previous values and instead the scores hover around the mean since ca 1600 CE. As noted, the trend here may be more evident if we standardize the axes in the figure.

P.10, Line 14 – This point is quite useful and we will add several sentences here that address a) relevant modern or historic patterns and b) other local factors (such as elevation) that may have contributed.

P. 11-12, Section 4.3 – In re-reading the text, we agree with the reviewer that this section is an ideal place to insert some more explicit comparison to modern patterns and to discuss the millennial-scale differences observed in each region (and how those differences may related to important processes or climate dynamics). We focused on the Pacific Northwest versus the Southwest because, here, in these two regions, we found the strongest case for a parallel to modern patterns caused by shifts in the position of the jet stream. However, coherent patterns elsewhere, especially in Central America and the northeast U.S. deserve further discussion. We can build upon work within these areas by previous studies of this interval (e.g., Hodell et al., 2005; Marlon et al., 2016). Likewise the PCAs and EOFs detect important differences between millennia in the mid-continent where we can draw on several decades of paleohydrologic work (e.g., Fritz et al., 2000) and analyses of historic droughts and floods (e.g., Schubert et al., 2004).

Table 1 – Where possible, we will obtain additional information about the age control of the various records.

Figure 1 – We can improve upon the figure design for clarity. The regional PCAs were excluded, as mentioned above, for regions with few records.

Figure 2 – The suggestions are good ones and will try to implement them.

Figure 4 – We can add histograms of values to go with each map.

References cited above:

Cayan, D. R.: Interannual climate variability and snowpack in the western United States, Journal of Climate, 9, 928–948, 1996.

Commission for Environmental Cooperation Working Group: Ecological Regions of North America - toward a common perspective, Commission for Environmental Cooperation, Montreal, Canada. [online] Available from: http://www3.cec.org/islandora/en/item/1701-ecological-regions-north-america-toward-common-perspective/ (Accessed 6 September 2017), 1997.

Fritz, S. C., Ito, E., Yu, Z., Laird, K. R. and Engstrom, D.: Hydrologic variation in the northern Great Plains during the last two millennia, Quaternary Research, 53, 175–184, 2000.

Hodell, D. A., Brenner, M., Curtis, J. H., Medina-González, R., Ildefonso-Chan Can, E., Albornaz-Pat, A. and Guilderson, T. P.: Climate change on the Yucatan Peninsula during the Little Ice Age, Quaternary Research, 63(2), doi:10.1016/j.yqres.2004.11.004, 2005.

Marlon, J. R., Pederson, N., Nolan, C., Goring, S., Shuman, B., Booth, R., Bartlein, P. J., Berke, M. A., Clifford, M., Cook, E., Dieffenbacher-Krall, A., Dietze, M. C., Hessl, A., Hubeny, J. B., Jackson, S. T., Marsicek, J., McLachlan, J., Mock, C. J., Moore, D. J. P., Nichols, J., Robertson, A., Schaefer, K., Trouet, V., Umbanhowar, C., Williams, J. W. and Yu, Z.: Climatic history of the northeastern United States during the past 3000 years, Clim. Past Discuss., 2016, 1–38, doi:10.5194/cp-2016-104, 2016.

Marsicek, J. P., Shuman, B., Brewer, S., Foster, D. R. and Oswald, W. W.: Moisture and temperature changes associated with the mid-Holocene Tsuga decline in the northeastern United States, Quaternary Science Reviews, 80, 129–142, doi:10.1016/j.quascirev.2013.09.001, 2013.

McKay, N. P.: A novel multiproxy approach: the PAGES North America 2k working group., PAGES magazine, 22, 100, 2014.

Mock, C. J.: Climatic Controls and Spatial Variations of Precipitation in the Western United States, Journal of Climate, 9, 1111, 1996.

Schubert, S. D., Suarez, M. J., Pegion, P. J., Koster, R. D. and Bacmeister, J. T.: On the Cause of the 1930s Dust Bowl, Science, 303(5665), 1855–1859, 2004.

Wise, E. K. and Dannenberg, M. P.: Persistence of pressure patterns over North America and the North Pacific since AD 1500, Nature Communications, 5, 4912, doi:10.1038/ncomms5912, 2014.

---

## Author Response (AR1)

Point-by-Point Response to reviews, Shuman et al., **cp-2017-35**
The text below provides our submitted responses to the reviews as well as bullets
listing the changes made to the manuscript.

The manuscript with the changes highlighted using 'Track Changes' follows.

Reviewer #1

The reviewer provides several insightful points that we will use to improve the
manuscript. As context, we would summarize our findings as evidence that long-
term (millennial) hydroclimate trends affected North America during the Common
Era, but that these trends were potentially small compared to both interannual
variation and Holocene-length trends. The records and attendant spatial patterns
are noisy because the signals of these trends and other centennial variations are
small compared to the sensitivity of many archives used to study Holocene-scale
variations. The points relate to three themes raised by the reviewer:

**Regions and regional patterns:** We agree that the spatial patterns do not
consistently express strong spatial coherence. We, therefore, propose revisions to
the text that explain the basis for the a priori regional assignments as well as
additional discussion of the limitations of our dataset, which could produce noisy
patterns (see also response to comment by S. St. George).

The regions were identified before our analysis as described by in a PAGES
workshop report by McKay (2014, p. 100): "Based on the dominant airmasses,
ecology, and the availability of proxy data throughout the continent, the group
developed initial spatial targets for subcontinental… reconstructions." However, the
distribution of data led to further divisions of the mid-continent and southern plains
as well as the northeast and southeast US to ensure that geographically disparate
locations were not averaged together; these decisions were guided by previously-
published modern climatological pattern analyses (Mock, 1996) before analysis of
the paleoclimate trends.

We propose to revise our text to more clearly explain the basis for the regions in
section 2.1.1. We propose to insert the text along the lines of the following after the
first sentence of section 2.1.1.:
"The nine regions were determined before we compiled the data considered here
(McKay, 2014), and were based on the level 1 ecoregions of North America
(Commission for Environmental Cooperation Working Group, 1997) and major
patterns of covariance within modern climate data (Mock, 1996). Where spatial
outliers existed separate from the main cluster of data within a region (e.g., data
from Florida versus the northeast U.S.; northern versus southern Great Plains), we
split our initial regions to ensure suitable representation of the data in our analysis.
We used these designations to ask whether distinct trends were recognizable

among commonly recognized regions and whether any trends have parallels to patterns of climate variation observed at finer time scales, such as north-south anti-phased moisture variability along the western margin of North America (Cayan, 1996; Wise and Dannenberg, 2014)."

- **We added the proposed text.**

In this way, we saw the analysis as related to modern climate patterns, although not as explicitly as raised by the reviewer. We propose to expand the Discussion section 4.1 to address potential parallels to historic patterns of variability within the noisy spatial patterns reconstructed by our analysis.

- **We have added text in Discussion section 4.1 to draw parallels to historic patterns. Please see p. 12, lines 6-20.**

We understand the questions about the PCAs for each region, but would clarify that the analyses served to help us examine intra-regional coherency as well as inter-regional correlation. Figure 2 shows both the temporal and spatial dimensions of patterns regardless of our regional designations, whereas Figure 5 attempts to assess how much variance is shared (based on PCs) among records within each region. Therefore, in section 2.2.3, we proposed to add:
"The PCA-by-region analysis was conducted to evaluate the strength of any signals within each region, rather than simply calculating mean trends, and to assess potential correlations or shared signals across geographically distinct regions. The EOF analysis evaluates the latter from the perspective of the whole dataset."

- **Added the proposed text.**

**Trends in the calibrated datasets:** The reviewer raises an excellent point about the power of the calibrated datasets. We included Figure 7 [now Fig. 8] as a means to demonstrate that the primary signal in these data is the dominant wetting trend. Unfortunately, these records are spatially clustered (primarily from the northeast U.S.) and may not be representative of the whole. Previous and forthcoming work has shown that no distinct differences exist between the pollen-inferred precipitation changes and P-E changes estimated from lake volume changes (Marlon et al., 2016; Marsicek et al., 2013). We propose to add these points to the text where suitable.

- **Related text was added on p. 8 between lines 8 and 17 in section 3.6.**

**Criteria for data inclusion and overall goals:** We had intended this manuscript to focus on the patterns observed in archives that are used to study Holocene-scale climate variations, and thus provide a bridge between the excellent work with dendroclimate records and studies of the whole interglacial. Therefore, we focused on archives that act as low-pass filters on paleoclimate changes. We agree that many different archives from tree rings to lake sediments may retain low-frequency signals or that such signals could derive from the characteristics of high-frequency events, and that the inclusion of some annually layered records appears to create an inconsistent application of criteria regarding what data were examined.

We should clarify, however, that our primary goal was to determine the signals captured by the types of records used to study the Holocene. We admit that we did not make this aim clear, but propose changing our title to emphasize this aspect of the study (see also our response to Review #2). This goal inherently led us to include ice cores, especially because they represent nearly direct measurements of past precipitation. Tree ring records have been the dominant source of past hydroclimate information over the Common Era, and in many ways, our analysis was intended not as a critique of that excellent work, but rather as a complement intended to ask: what patterns of change are recorded by "everything else"?

- **We have revised the text in the Introduction, particularly in the third paragraph to make our goals and data selection clear**.
- **We have also changed the title to help clarify the focus.**

Specific comments:

P. 3 Line 16 – We agree that we need to standardize the wording for clarity about hydroclimate versus hydrologic changes, and replace the word "calibrate" in this particular line with "interpreted to represent a specific climate variable."

- **We have tried to use the term, hydroclimate, as consistently as possible.**

P. 5 Line 25 – We will provide a sentence clarifying the application of SSA to missing values.

- **We have added explanatory text to the end of section 2.2.1.**

P. 6 Line 14 – As noted above, we propose to clarify about the regions. PCA was not applied where only one or two sites exist. Only the regions shown in Fig. 5 with >10 records were analyzed. We will add text to explain this point.

- **Added relevant text as two new sentences at the end of section 2.2.3.**

P. 6 Line 18 – "Interferences" was a typo and should be "inferences".

- **Corrected the spelling.**

P. 7, Line 2 – We will rephrase to ensure that the references to the clusters in Figure 2 are clear. The suggestion to refer to them by color seems like it may be useful.

- **In section 3.1 we have cited the color used to indicate each cluster in Figure 2.**

P. 7, Line 5 – We can see the trend that the reviewer mentions and will try to describe it. However, the primary feature identified as a deviation for the mean in the EOF is the Medieval Anomaly. The EOF contrasts the early portion of these records with the Medieval Period, but we agree that a background trend also exists because the records never return fully to their previous values and instead the scores hover around the mean since ca 1600 CE. As noted, the trend here may be more evident if we standardize the axes in the figure.

- **We revised the second paragraph of section 3.1 and standardized the axes in the figure.**

P.10, Line 14 – This point is quite useful and we will add several sentences here that address a) relevant modern or historic patterns and b) other local factors (such as elevation) that may have contributed.
- **We added several lines about comparisons to other patterns and sources of heterogeneity to the second paragraph of section 3.1.**

P. 11-12, Section 4.3 – In re-reading the text, we agree with the reviewer that this section is an ideal place to insert some more explicit comparison to modern patterns and to discuss the millennial-scale differences observed in each region (and how those differences may related to important processes or climate dynamics). We focused on the Pacific Northwest versus the Southwest because, here, in these two regions, we found the strongest case for a parallel to modern patterns caused by shifts in the position of the jet stream. However, coherent patterns elsewhere, especially in Central America and the northeast U.S. deserve further discussion. We can build upon work within these areas by previous studies of this interval (e.g., Hodell et al., 2005; Marlon et al., 2016). Likewise the PCAs and EOFs detect important differences between millennia in the mid-continent where we can draw on several decades of paleohydrologic work (e.g., Fritz et al., 2000) and analyses of historic droughts and floods (e.g., Schubert et al., 2004).
- **We have added paragraphs summarizing changes in central America and the northeast U.S.**

Table 1 – Where possible, we will obtain additional information about the age control of the various records.
- **We have provided URLs to the data within the NOAA Paleoclimate Data Center.**

Figure 1 – We can improve upon the figure design for clarity. The regional PCAs were excluded, as mentioned above, for regions with few records.
- **Figure 1 has been re-plotted for clarity.**

Figure 2 – The suggestions are good ones and will try to implement them.
- **We standardized the axes and used a different map projection to improve Fig. 2**

Figure 4 – We can add histograms of values to go with each map.
- **This is now Fig. 5 and we have added the histograms as suggested as well as North America Drought Atlas reconstructions for comparison.**

Reviewer #2

The reviewer raises three good points, which we would propose to address through revision.

**Comparison to tree-ring reconstructions**: Both reviews as well as St. George's comment highlight that the comparison is the obvious one to make. We struggled with this topic in designing the manuscript. Therefore, we propose to 1) clarify our goals for the paper, including by changing the title, and 2) include and discuss a comparison with patterns in the North American drought atlas.

In the first case, as noted in our response to review 1, we intended to evaluate the patterns that exist as recorded by data other than the excellent dendroclimatic record. We intended such an analysis to focus on patterns at multi-centennial to millennial scales, in part, as a way to place the Common Era in the context of the whole of the Holocene. We, therefore, propose changing our title to read: "Placing the Common Era in a Holocene Context: Millennial-to-centennial patterns and trends in the hydroclimate of North America over the past 2000 years."
- **We have revised the title.**

Not all of the available data were suitable for providing a Holocene context (because of limited time depth), but the dataset was aimed at methods and archives that could capture the relevant variations. They represent the methods used to study Holocene and even Pleistocene changes – and a useful question is whether they have the sensitivity to always record climate variation within just two millennia or less as tree rings can clearly do. New methods in dendroclimatology certainly preserve many low-frequency patterns, but these methods have not been applied to all reconstructions and only cover certain regions. Including the high number of tree-ring chronologies would have also dominated the patterns in our analysis for some areas and prevented an independent perspective afforded by the other (Holocene-scale) data types.
- **We have revised the Introduction to make related points.**

Therefore, we aimed to be independent of this major data source (dendroclimate data), but we agree, it begs then for comparison. So, toward, the second proposed revision, we can include maps of PDSI anomalies from the North American Drought Atlas (NADA) as a point of comparison in Fig. 4 [Now Fig. 5]. Our preliminary assessment indicates a poor correlation, but we agree even a mismatch is worthy of more discussion: why does it exist? We would propose to revise our text to discuss the potential reasons.
- **In the revised Fig. 5, we have added the NADA results for comparison.**

We would add text similar to the following as a new sub-section in the Discussion (section 4):

- **We have added the following text as proposed as a new section at the end of the Discussion.**

"Differences between the patterns in our dataset and the NADA may exist for several reasons. First, contrasts may exist between the way our dataset retains signals of annual-to-decadal variations (clearly preserved in the dendroclimate record) and those of multi-century and longer variations. For example slow sediment dynamics or forest tree longevity may prove resilient to annual variations, but readily responsive to change over centuries. Different map patterns could arise, therefore, because NADA likely emphasizes interannual variation, even when smoothed over centuries, whereas other datasets may emphasize the effects of centennial and longer changes. The differences would represent different patterns in the averages of high-frequency variability versus the patterns in low-frequency trends.

Furthermore, our dataset may lack a consistent ability to detect either annual-decadal variability or multi-century trends in a limited 2000-yr window because of the interaction of taphonomic process (e.g., sediment mixing) and the small magnitudes of the low-frequency trends. The datasets may well be noisy relative to weak low-frequency signals. As we noted in the Introduction, the magnitudes of the trends in many records are small over even 2000 yrs (Fig. 7) when compared to many reconstruction uncertainties. The low signal-to-noise ratio may also apply to the dendroclimate data at multi-century to millennial scales, and without long observational datasets available for validation, it is difficult to assess.

A third related explanation for mismatches could be that dendroclimatic reconstructions of variables such as the Palmer Drought Severity Index (PDSI) may differ from the hydroclimate variables represented by our data (e.g., net snow accumulation in ice cores; P-ET that drives lake-level changes). We do recognize some clusters of coherent anomalies (e.g., clusters of opposite sign anomalies in the Pacific Northwest versus the U.S. Southwest in Fig. 4), which would at first pass, suggest real signal and thus, in part, require explanations that involve differences in the time scale (explanation 1) or controlling variable (explanation 3) recorded by the two different datasets. More work is needed to test the various explanations."

Overall, we agree that lessons probably lie hidden within even mismatched patterns and that we should show the comparison.

**Selection of time windows:** We would propose to add text to section 2.2.1 that clarifies that we used 100-yr bins as a simple approach to focusing on trends that were longer than a century. Some data do have the ability to assess finer patterns of variation, but we will clarify that our focus was on the low-frequency aspects of the records. Additionally, our focus on the last 2000 years was intended to align with the goals of PAGES 2k. Other timeframes may have been chosen to maximize the available data, but 2000 yrs was also a useful contrast with Holocene trends, which we intended as a focus of the manuscript. Similar to the suggestion about sub-setting the data, we split our data into 2ka and Holocene length groups to examine,

first, the dominant patterns (if any) over the last 2 ka in these types of records and, then, how these patterns compared with those that extend through the Holocene.

- **We added a sentence about 100-yr bins at the end of the first paragraph of section 2.1.1.**
- **We have provided a new figure showing the data split by archive type and added a description of these trends as a new paragraph at the end of section 3.1.1.**

Our maps contrasting the two millennia were designed with two points in mind, which we would propose to clarify. First, what were the spatial patterns of the long-term trends represented by the EOFs and PCs? Second, were differences observed when we contrast the Medieval and Little Ice Age periods a function of the long trends? However, the reviewer raises useful points and caveats.

- **We have added a related sentence to section 2.2.4.**

**Data Groups**: The point about evaluating the data by type is a good one and we would propose to include a figure showing the mean patterns or PCAs by proxy. Breaking the data apart in this way may help elucidate the causes of different patterns in our dataset and NADA.

- **As noted above, we added a related revised figure and text in section 3.1.1.**

Short comments by Scott St. George

Response 1:
Thank you to the review team for their suggestions. We agree to lodge the specified datasets with a public repository, as indicated in the "data availability" section.

Response 2:
We agree that all of the datasets involved have both strengths and limitations. We propose to add brief discussions of limitations to sections 2.1.2-2.1.5 where we describe the different archives used in our analysis. We appreciate the reference to the Huybers et al. paper. It is a good point that "low frequency" variation in systems with autoregressive memories of various durations can arise both from low frequency and high frequency climate forcing, and that it is not always clear which factor (or their interaction) is important for a given record.

- **We have revised the third paragraph of the Introduction to explicitly mention weaknesses of the data that we examined here.**
- **We have added a sentence about the contributions of low and high frequency changes, and autoregressive memories, in the Introduction, fourth paragraph.**

We had not intended our comments or analysis as a critique of dendroclimate records, but we can see that our wording did not fully clarify our intention to evaluate the patterns in the types of data normally used to study long-term changes extending through the Holocene. Please see our related responses to the two reviews and our proposed title revision: "Placing the Common Era in a Holocene Context: Millennial-to-centennial patterns and trends in the hydroclimate of North America over the past 2000 years."

- **We reduced some of our discussion of the limitations of dendroclimate records in the Abstract and Introduction.**
- **We have modified the title as proposed and shifted the Introduction to focus on the comparison to Holocene trends.**

[revised manuscript text omitted]

---

## Author Response (AR2)

We are excited about the decision to accept the manuscript for publication and appreciate the need to wait for all of the relevant data to have been submitted and available on-line.

Toward that end, we have now submitted all of the datasets used in our study to the NOAA Paleoclimate archive and have updated the related URLs in Table 1 to represent the new file locations. We expect that all of the URLs will be live soon.

The text in the manuscript related to data availability has been updated accordingly.

Thank you for the editorial support and guidance. Please let us know about any additional changes that should be made.